# AI4SLT: Empirical Processes in Lean 4 for Formal Statistical Learning Theory

Yuanhe Zhang [1]   Jason D. Lee [2]   Fanghui Liu [3]

## Abstract

We present the first comprehensive Lean 4 formalization of statistical learning theory (SLT) grounded in empirical process theory. Our end-to-end formal infrastructure implement the missing contents in latest Lean library, including a complete development of Gaussian Lipschitz concentration, Dudley's entropy integral theorem for sub-Gaussian processes, and an application to least-squares (sparse) regression with a sharp rate. The project was carried out using a human-AI collaborative workflow, in which humans design proof strategies and AI agents execute tactical proof construction, leading to the human-verified Lean 4 toolbox for SLT. Beyond implementation, the formalization process exposes and resolves implicit assumptions and missing details in standard SLT textbooks, enforcing a granular, line-by-line understanding of the theory. This work establishes a reusable formal foundation and opens the door for future developments in machine learning theory. The code is provided in https://github.com/YuanheZ/lean-stat-learning-theory.

## 1. Introduction

Statistical learning theory (SLT), and more generally, machine learning theory, successfully guided the progress of machine learning over the past two decades, informing foundational concepts such as bias-variance trade-offs, regularization, and cross-validation (Hastie et al., 2009). Now it tries to capture the picture for complex architectures such as deep neural networks (LeCun et al., 2015) and large language models (Brown et al., 2020) at some points, e.g., double descent (Belkin et al., 2019; Mei & Montanari, 2022), benign overfitting (Bartlett et al., 2020; Tsigler & Bartlett, 2023), and single/multi-index model (Montanari & Urbani, 2025; Abbe et al., 2022; Bruna & Hsu, 2025).

However, as models become increasingly complex, contemporary theoretical analyses have grown substantially longer and more intricate. Modern proofs often rely on a wide range of advanced mathematical tools or inspired by statistical physics. This broad techniques place significant strain on human review (i.e., ***verification at scale***): it becomes difficult to verify intermediate lemmas, track logical dependencies, and clearly identify which techniques are applicable at each stage of the argument. Besides, some core techniques in SLT, e.g., concentration inequalities, covering, are without a structured, machine-readable library, leading to ***untapped reusability***.

Formalization in interactive theorem provers such as Lean 4 (Moura & Ullrich, 2021) addresses both challenges and is rapidly gaining traction in the math community. By encoding proofs in a formal language, we obtain machine-checkable correctness guarantees while simultaneously creating a structured, queryable library of results. We argue that formalizing SLT is not merely an exercise in rigor, but a foundation for scalable, automated theoretical analysis of machine learning systems. Current Lean 4 implementation in machine learning includes reinforcement learning theory (Zhang, 2025), optimization (Li et al., 2024; 2025a;b). The most close to our work is conducted by Sonoda et al. (2025) on generalization bounds via Rademacher complexity, but limited to simple settings, see the discussion in Section 2.2.

Unlike more self-contained mathematical areas such as number theory or algebra, where formalization has flourished thanks to clean axiomatic foundations, SLT lies at the intersection of multiple disciplines, related to empirical process theory (Van Der Vaart & Wellner, 1996). To be specific, as shown in Figure 1, the excess risk of a learning algorithm is governed by the supremum of an empirical process indexed by the loss class. Controlling this supremum requires two interlocking components: *concentration inequalities* (Boucheron et al., 2013) that convert high-probability bounds into link to complexity measure, and *capacity control* that quantifies the effective size of localized function

---

[1]Department of Statistics, University of Warwick, UK (*yuanhe.zhang@warwick.ac.uk*). [2]EECS and Statistics, University of California, Berkeley, USA (*jasondlee88@gmail.com*). [3]School of Mathematical Sciences, Institute of Natural Sciences and MOE-LSC, Shanghai Jiao Tong University, China. Part of work was done at Department of Computer Science, University of Warwick, UK. Correspondence to: Fanghui Liu <fanghui.liu@sjtu.edu.cn>.

*Proceedings of the $43^{rd}$ International Conference on Machine Learning*, Seoul, South Korea. PMLR 306, 2026. Copyright 2026 by the author(s).

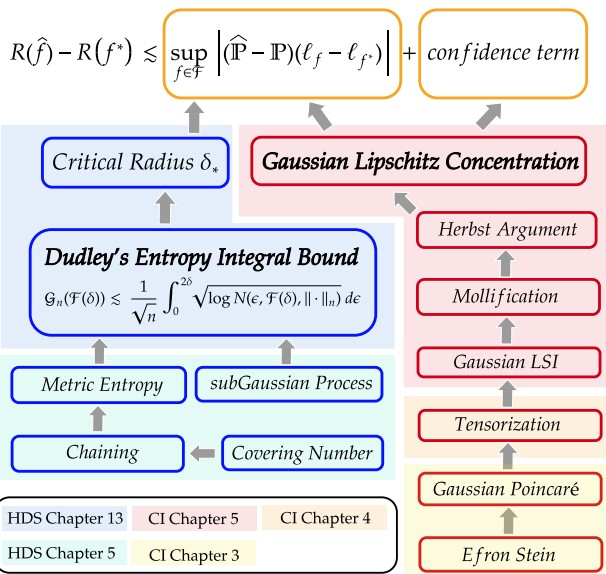

$$R(\hat{f}) - R(f^*) \lesssim \sup_{f \in \mathcal{F}} \left| (\hat{\mathbb{P}} - \mathbb{P})(\ell_f - \ell_{f^*}) \right| + confidence\ term$$

*Figure 1.* Lean formulation for Localized Empirical Process Framework. It includes the blue part for the capacity control and the red part for concentration. The colored zone indicates the major results in the chapters of Wainwright (2019) (High Dimensional Statistics, HDS) and Boucheron et al. (2013) (Concentration Inequality, CI).

classes via complexity measure and metric entropy. Each tool demands careful treatment of measurability, integrability, and topological assumptions that textbooks routinely leave implicit. More importantly, these tools remain extremely **undeveloped** in Lean 4.

In this work, we rise to this challenge by formalizing all infrastructures for SLT in Lean 4 *from scratch* via a human–AI collaboration system. Starting from basic measure-theoretic probability and analysis, we systematically develop the full stack of tools required for modern generalization analysis. The dependency structure, illustrated in Figure 1, reveals our formalization path, including several key parts of representative books (Wainwright, 2019; Boucheron et al., 2013). Our contributions are:

**1. Implementation on Gaussian Lipschitz Concentration:** We construct a *complete formal development of Gaussian Lipschitz concentration* which requires building a substantial infrastructure across Efron-Stein inequality, Gaussian Poincaré inequality, Density argument, and Gaussian logarithmic Sobolev inequality (LSI). The Gaussian LSI, in particular, is a foundational tool in high-dimensional probability with far-reaching applications beyond learning theory. To our knowledge, this constitutes *the first formalization of the complete Gaussian analysis tools* in any theorem prover.

**2. Implementation on Dudley's Entropy Integral:** We provide the *first formalization of Dudley's entropy integral theorem for sub-Gaussian processes* in Lean 4. This is a cornerstone result in empirical process theory that bounds the

expected supremum of a stochastic process by an integral involving metric entropy. Our formalization encompasses the full generality of sub-Gaussian processes. The development required formalizing the sophisticated chaining technique that decomposes a stochastic process into a telescoping sum over dyadic approximations, along with rigorous treatment of covering and packing numbers in metric spaces.

**3. Application: Least-Squares Framework via Localized Empirical Process:** We demonstrate the practical utility of our formalizations by developing a unified framework for least-squares regression based on the localized empirical process. We further test the functionability of our formal unified framework on linear regression and $\ell_1$-constrained regression to obtain sharp rates up to minimax-level.

**4. Human-AI Collaborative Formalization Paradigm:** Our formalization is completed through structured collaboration between human mathematicians and Claude Code (Anthropic, 2025a) with Opus-4.5 (Anthropic, 2025b). Humans analyze Mathlib's infrastructure, design proof strategies, and decompose complex theorems into manageable lemmas; the AI agent executes these plans and constructs formal proofs. The entire process totals ~500 hours of supervised development, with all formalizations compiled without `sorry` or `axiom`. This provides one *realization* of how large-scale formalization projects can be substantially accelerated through well-designed human–AI collaboration.

The scale of our contribution is substantial: the project comprises approximately 30,000 lines of Lean 4 code. We provide a list of our major formalizations in Section A. Crucially, this effort **goes far beyond implementation**. Achieving a complete formalization requires a granular, line-by-line understanding of SLT: every definition, assumption, inequality, and logical dependency must be explicitly identified, verified, and composed into a coherent proof structure.

This makes the project particularly valuable for student training in theoretical machine learning. Engaging with the formalization demands mastery of the full technical stack of SLT rather than superficial familiarity. By providing a rigorous, end-to-end formal infrastructure for SLT grounded in the empirical process, this work offers a principled training ground and opens the door for students seeking to develop deep theoretical competence in modern machine learning.

## 2. SLT from Natural Language to Lean 4

In this section, we take an overview of the structure of SLT in natural language and diagnose what are missing or should be built in Lean 4.

### 2.1. Statistical learning theory for generalization

Empirical process theory provides a unified uniform convergence framework for generalization guarantees of learning

algorithms by exploiting the geometry of function classes.

As shown in Figure 1, formally, let $\mathcal{F}$ be a hypothesis class and $\ell_f(z)$ be a loss function associated with $f \in \mathcal{F}$. For a broad class of empirical risk minimization and regularized learning procedures, the excess risk of an estimator $\hat{f}$ admits

$$R(\hat{f}) - R(f^\star) \lesssim \underbrace{\sup_{f \in \mathcal{F}} \left| (\hat{\mathbb{P}} - \mathbb{P})(\ell_f - \ell_{f^\star}) \right|}_{\text{empirical process fluctuation}} + (\text{confidence}),$$

where $\mathbb{P}$ and $\hat{\mathbb{P}}$ denote the population and empirical measures, respectively. This decomposition highlights that generalization is governed by the uniform deviation of an empirical process indexed by the excess loss class. The global structure of this framework includes two main parts via several probabilistic toolbox (see Figure 1).

**Concentration:** High-probability bounds on the empirical process fluctuation can be obtained via concentration inequalities, such as *Gaussian Lipschitz concentration* (Wainwright, 2019, Corollary 14.15). These yield bounds in the form of *critical radius* $\delta_\star$.

**Capacity Control:** The key to obtain $\delta_\star$ is *localization*: rather than controlling the empirical process over the entire class $\mathcal{F}$, one restricts attention to a localized class $\mathcal{F}(\delta) = \{f \in \mathcal{F} : d(f, f^\star) \leq \delta\}$ consisting of functions within radius $\delta$ of the optimum with respect to a suitable metric $d$.

The effective size of $\mathcal{F}(\delta)$ is quantified by localized complexity measures, e.g., Gaussian complexity $\mathcal{G}_n(\mathcal{F}(\delta))$ and Rademacher complexity $\mathcal{R}_n(\mathcal{F}(\delta))$. Their connection to geometry is made explicit through *metric entropy*, $\log N(\mathcal{F}(\delta), \epsilon, d)$ defined via *covering numbers*: how many balls of radius $\epsilon$ are needed to cover $\mathcal{F}(\delta)$. In particular, *chaining arguments*, most notably Dudley's entropy integral, relate localized Gaussian complexity to metric entropy:

$$\mathcal{G}_n(\mathcal{F}(\delta)) \lesssim \frac{1}{\sqrt{n}} \int_0^{2\delta} \sqrt{\log N(\epsilon, \mathcal{F}(\delta), d)} \, d\epsilon \,.$$

This characterization reveals how complexity accumulates across scales and leads to a solution set whose smallest solution defines the critical radius $\delta_\star$. The resulting bounds recover sharp minimax rates in parametric settings and extend naturally to nonparametric models.

### 2.2. Formalization Gaps

Despite the theoretical maturity of the above framework, its formalization in Lean 4 remains in its infancy. Recent work by Sonoda et al. (2025) formalizes generalization bounds via Rademacher complexity, including basic tools such as McDiarmid's inequality and Hoeffding's lemma. However, their analysis controls the empirical process over the **entire** function class, leading to loose rates and limited applica-

tions. The **sharper localized empirical process framework** goes beyond this in two folds:

First, the required **concentration machinery** in this project is substantially more advanced. While McDiarmid-type inequalities suffice under boundedness assumptions, localized analysis relies on Gaussian Lipschitz concentration, whose proof draws on a deep chain of results from functional analysis and probability theory, that requires formulation or significant changes in Lean 4.

Second, the capacity control has quite limited formulation in latest Lean library. Formalizing localization requires developing covering numbers, chaining arguments underlying Dudley's integral, localized complexity measures, and the fixed-point analysis that determines the critical radius.

Our work bridges precisely this gap via the comprehensive Lean 4 formalization. Importantly, this effort ***goes far beyond mechanical translation***. Natural-language proofs routinely suppress measurability and topological assumptions, conflate almost-sure and pointwise statements, and compress multi-step arguments into informal phrases. Formalization forces each of these gaps to be made explicit and resolved. Moreover, Lean demands careful proof engineering: for example, formulazation of Dudley's entropy integral requires systematic coordination between different notions of integration (e.g., Bochner and interval integrals), which are distinct in Lean but mixed in language proofs.

## 3. Formulation details and challenges

In this section, we present the details of our formalization (see Figure 2), covering Gaussian Lipschitz concentration in Section 3.1 and Dudley's entropy integral bound in Section 3.2. For clarity, we first state each result in natural-language theorem form, followed by the corresponding Lean 4 formalization, where we explicitly discuss the key modeling and proof-engineering challenges.

### 3.1. High-Dimensional Gaussian Analysis Toolbox

Gaussian functional inequalities are the backbone of HDS and SLT. However, the complete proofs form a long chain of disparate methods. Each link relies on a different piece of analysis, and each is nontrivial to formalize. We build a reusable, deliberately end-to-end formal toolbox that supports the full analytic pipeline from scratch, see the red part of Figure 2 with the following steps.

**Theorem 3.1** (**i. Efron-Stein's Inequality**, Theorem 3.20 in Boucheron et al. (2013))**.** *Let* $\boldsymbol{X} = (X_1, \dots, X_n)$ *be a vector of* $n$ *independent random variables and* $Z = f(\boldsymbol{X})$ *be a square-integrable function of $X$. Denote $E^{(i)}$ as the conditional expectation conditioned on*

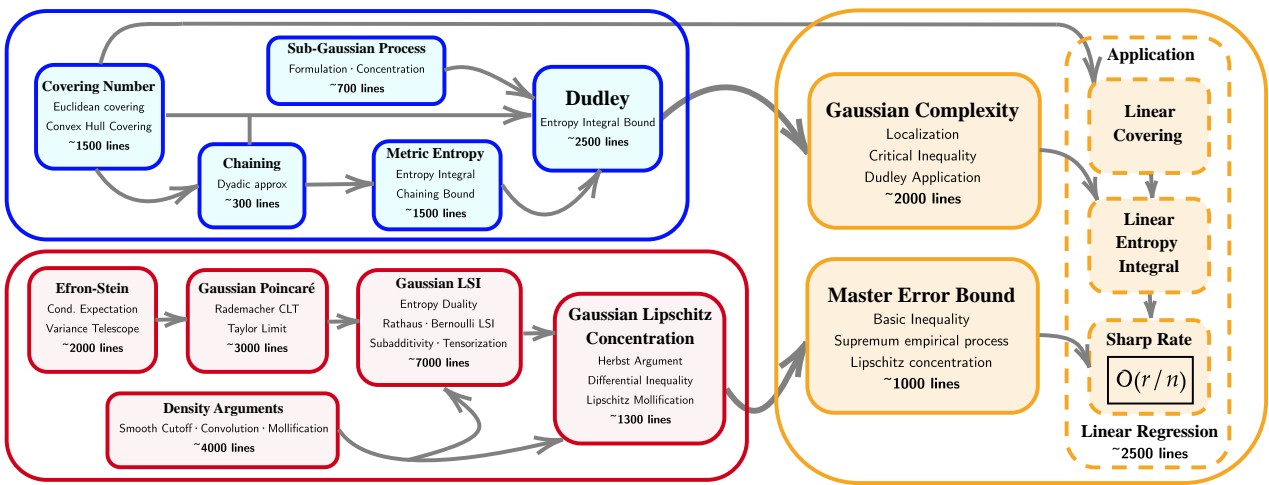

*Figure 2.* The dependency graph of our formalizations. All the contents in the graph have not been implemented in Lean 4 before.

$(X_1, ..., X_{i-1}, X_{i+1}, ..., X_n)$. *Then,*

$$\mathrm{Var}(Z) \leq \sum_{i=1}^{n} \mathbb{E}\left[\left(Z - \mathbb{E}^{(i)}[Z]\right)^2\right].$$

We start with formalizing $E^{(i)}$ as follows:

```
noncomputable def condExpExceptCoord (i :
    Fin n) (f : (Fin n → Ω) → ℝ) : (Fin
    n → Ω) → ℝ :=
  fun x => ∫ y, f (Function.update x i y) ∂
    (μs i)
```

**Implementation challenge:** Theorem 3.1 allows each $X_i$ to have a distinct distribution $\mu_i$, which complicates measure-theoretic arguments for coordinate-wise updates. We address this by formalizing a universal transfer lemma: re-sampling one coordinate with a fresh independent sample preserves the joint distribution. Though requiring extra formalization of measure-theoretic machinery (measure rectangles), this lemma powers 20+ usages across tower properties, Fubini-style swapping, and slice integrability.

Then, we can formalize Theorem 3.1 as:

```
theorem efronStein (f : (Fin n → Ω) → ℝ)
    (hf : MemLp f 2 μˢ) :
    variance f μˢ ≤ Σ i : Fin n, ∫ x, (f x
    - condExpExceptCoord (μs := μs) i f
    x)^2 ∂μˢ := by
```

**ii. Gaussian Poincaré Inequality:** To formalize Gaussian Poincaré inequality, we need to use Efron-Stein's infrastructures and require a series of results as below.

**Corollary 3.2.** *Let $X$ be a standard Gaussian random variable and $f \in C_c^\infty(\mathbb{R})$. Then, $\mathrm{Var}[f(X)] \leq \mathbb{E}\left[f'(X)^2\right]$.*

Notice that Corollary 3.2 is not directly used to derive the Gaussian LSI, instead its intermediate proof is re-used.

**Implementation challenge:** Formalizing the Corollary 3.2 combines Taylor expansion bounds with weak convergence of Rademacher sums to Gaussian, which needs careful measure-theoretic tracking through bounded continuous function wrappers and coordinate-permutation symmetry. Notice that such machinery is frequent in Gaussian analysis.

We formalize Corollary 3.2 as:

```
theorem gaussianPoincare {f : ℝ → ℝ} (hf
    : CompactlySupportedSmooth f) :
    variance (fun x => f x)
    stdGaussian.toMeasure ≤
    ∫ x, (deriv f x)^2 ∂
    stdGaussian.toMeasure := by
```

**iii. Density Arguments:** The density arguments provide an efficient tool which let people prove inequalities for smooth and compactly supported function class $C_c^\infty$ (easier to apply convergence theorems) then extend to general class by such argument. This is the key to: 1) extend Gaussian LSI from $C_c^\infty$ to $C^1$, and 2) extend Gaussian Lipschitz concentration from $C_c^\infty$ to the general Lipschitz class. Such arguments are often **_skipped in textbooks_** (Boucheron et al., 2013) due to the complexity. We start with defining the membership of Gaussian Sobolev space[1] $\mathcal{W}^{1,2}(\gamma^{\otimes n})$ as:

```
def MemW12Gaussian (n : ℕ) (f : E n → ℝ)
    (γ : Measure (E n)) : Prop :=
  MemLp f 2 γ ∧ MemLp (fun x ↦ fderiv ℝ f
    x) 2 γ
```

and the squared Gaussian Sobolev norm as:

---

[1]Here $f$ should be continuously differentiable which ensures the derivative is well-defined so we can use fderiv from Lean 4.

```
noncomputable def GaussianSobolevNormSq (n
    : ℕ) (f : E n → ℝ) (γ : Measure (E
    n)) : ℝ≥0∞ :=
  eLpNorm f 2 γ ^ 2 + eLpNorm (fun x ↦ ‖
    fderiv ℝ f x‖) 2 γ ^ 2
```

The main density theorem is provided by:

**Theorem 3.3.** *The space of smooth compactly supported functions $C_c^\infty$ is dense in $\mathcal{W}^{1,2}(\gamma^{\otimes n})$, where $\gamma$ is the standard Gaussian measure.*

This can be used to extend Gaussian LSI to $C^1$ class with the following Lean 4 formulation.

```
theorem
dense_smooth_compactSupport_W12Gaussian :
  ∀ f : E n → ℝ, MemW12Gaussian n f
    (stdGaussianE n) →
  Differentiable ℝ f →
  Continuous (fun x => fderiv ℝ f x) →
  ∀ ε > 0, ∃ g : E n → ℝ, ContDiff ℝ (⊤ :
    ℕ∞) g ∧ HasCompactSupport g ∧
    GaussianSobolevNormSq n (f - g)
    (stdGaussianE n) < ENNReal.ofReal ε :=
    by
```

**Remark:** `stdGaussianPi` is the product measure of $n$ independent standard Gaussians, with its pushforward `stdGaussianE` to Euclidean space via the equivalence.

To extend the concentration theorem, we need a specialized density lemma based on the Lipschitz mollification technique. In our formalization, we pick a nonnegative mollifier $\rho \in C_c^\infty(\mathbb{R}^n)$ with $\int \rho(\boldsymbol{x})\,d\boldsymbol{x} = 1$ then define a smooth approximation to Lipschitz function $f$ via

$$\rho_\epsilon(\boldsymbol{x}) = \epsilon^{-n}\rho(\boldsymbol{x}/\epsilon), \quad f_\epsilon := f * \rho_\epsilon. \tag{1}$$

Notice that $f_\epsilon$ lives within $C_c^\infty(\mathbb{R}^n)$ and preserves the Lipschitz constant of $f$. The lemma is presented as:

**Lemma 3.4.** *Let $f : \mathbb{R}^n \to \mathbb{R}$ be a Lipschitz function, there is a constant $C_\rho := \int \|\boldsymbol{u}\|_2 \rho(\boldsymbol{u})\,d\boldsymbol{u} < \infty$ such that*

$$\sup_{\boldsymbol{x} \in \mathbb{R}^n} |f_\epsilon(\boldsymbol{x}) - f(\boldsymbol{x})| \leq LC_\rho \epsilon.$$

*Hence, $f_\epsilon \to f$ uniformly as $\epsilon \downarrow 0$.*

Our formalization of Lemma 3.4 is:

```
theorem mollify_tendsto_of_lipschitz {f :
    E n → ℝ} {L : ℝ≥0} (hf :
    LipschitzWith L f) (x : E n) :
  Filter.Tendsto (fun ε => mollify ε f x)
    (nhdsWithin 0 (Set.Ioi 0)) (nhds (f
    x)) := by
```

**Implementation challenge:** We formalize a large amount of smooth approximation and convolutions in this part, which is an integration of functional analysis with measure-theoretic probability theory.

**Theorem 3.5** (**iv. Gaussian LSI**, Theorem 5.4 of (Boucheron et al., 2013)). *Let $\boldsymbol{X} = (X_1, \dots, X_n)$ be a vector of $n$ independent standard Gaussian random variables and $f : \mathbb{R}^n \to \mathbb{R}$ be a continuously differentiable function with $\mathbb{E}[f'(X)] < \infty$. Then, $\mathrm{Ent}(f^2) \leq 2\mathbb{E}\left[\|\nabla f(\boldsymbol{X})\|_2^2\right]$.*

By defining the entropy $\mathrm{Ent}(f)$ as

$$\mathrm{Ent}(f) = \mathbb{E}\left[f(X)\log f(X)\right] - \mathbb{E}[f(X)]\log\mathbb{E}[f(X)],$$

with its formulation,

```
def entropy (μ : Measure Ω) (f : Ω → ℝ) :
    ℝ :=
  ∫ ω, f ω * log (f ω) ∂μ - (∫ ω, f ω ∂μ)
    * log (∫ ω, f ω ∂μ)
```

now we formalize Theorem 3.5 as:

```
theorem gaussian_logSobolev_W12_pi {n : ℕ}
    {g : (Fin n → ℝ) → ℝ}
  (hg : MemW12GaussianPi n g
    (stdGaussianPi n)) (hg_diff :
    Differentiable ℝ g) (hg_grad_cont : ∀
    i, Continuous (fun x => partialDeriv i
    g x)) (hg_log_int : Integrable (fun x =
    > (g x)^2 * log ((g x)^2))
    (stdGaussianPi n)) :
  entropy (stdGaussianPi n) (fun x => (g
    x)^2) ≤ 2 * ∫ x, gradNormSq n g x ∂
    (stdGaussianPi n) := by
```

where `gradNormSq` is the squared norm of gradients.

In the next, we briefly present the high-level proof strategy for formalization. We first formalize one-dimensional case and generalize to dimension-free via tensorization later. Now let $\varepsilon_1, \dots, \varepsilon_n$ be $n$ independent Rademacher random variables and fix $f \in C_c^2(\mathbb{R})$. Define the Rademacher sum $S_n := n^{-1/2}\sum_{k=1}^n \varepsilon_k$, building upon the infrastructures in Theorem 3.1, we use Taylor's limit and CLT to obtain

$$\lim_{n\to\infty} \sum_{k=1}^n \mathbb{E}\left[f\left(S_n + \frac{1-\varepsilon_k}{\sqrt{n}}\right) - f\left(S_n - \frac{1+\varepsilon_k}{\sqrt{n}}\right)\right]^2$$
$$= 4\mathbb{E}\left[f'(X)^2\right], \quad X \sim \mathcal{N}(0,1). \tag{2}$$

For $f \in C_c^2(\mathbb{R})$, by CLT, we can obtain

$$\lim_{n\to\infty} \mathrm{Ent}\left[f^2(S_n)\right] = \mathrm{Ent}[f(X)^2]. \tag{3}$$

We then bridge Eq. (2) and Eq. (3) by formalizing the Bernoulli logarithmic Sobolev inequality (LSI) (Boucheron et al., 2013, Theorem 5.1) then taking limit to both sides, we can derive the 1D Gaussian LSI for $f \in C_c^2(\mathbb{R})$, i.e. $\mathrm{Ent}(f^2) \leq 2\mathbb{E}\left[f'(X)^2\right]$.

**Remark:** For the proof of Gaussian Lipschitz concentration, we can directly tensorize this $C_c^2(\mathbb{R})$ version to be dimension-free. We further use Theorem 3.3 to extend the above inequality from to $C^1(\mathbb{R})$ for a general toolbox.

Since Gaussian LSI is a direct consequence of entropy subadditivity and one-dimensional case of LSI, we need the formulation of the subadditivity of entropy theorem (Boucheron et al., 2013, Theorem 4.22), the key technique of tensorization.

**Theorem 3.6** (**subadditivity**, Theorem 4.22 of (Boucheron et al., 2013)). *Let $\boldsymbol{X} = (X_1, \ldots, X_n)$ be a vector of $n$ independent random variables and $Y = f(\boldsymbol{X})$ be a nonnegative measurable function of $\boldsymbol{X}$ such that $\Phi(Y) = Y \log Y$ is integrable. Define $\mathrm{Ent}^{(i)}(Y)$ as the conditional entropy given $(X_1, \ldots, X_{i-1}, X_{i+1}, \ldots, X_n)$. Then,*

$$\mathrm{Ent}(Y) \leq \mathbb{E}\left[\sum_{i=1}^{n} \mathrm{Ent}^{(i)}(Y)\right].$$

The formalization follows from similar telescoping strategy in Theorem 3.1 and our formalization of duality formula

$$\mathrm{Ent}(Y) = \sup_{T} \mathbb{E}\left[Y\left(\log T - \log \mathbb{E}(T)\right)\right],$$

where the supremum is over all integrable and nonnegative random variables.

```
theorem entropy_subadditive (f : (Fin n →
    Ω) → ℝ) (hf_meas : Measurable f)
    (hf_nn : 0 ≤ᵐ[μˢ] f) (hf_int :
    Integrable f μˢ) (hf_log_int :
    Integrable (fun x => f x * log (f x)) μˢ
    ) :
  LogSobolev.entropy μˢ f ≤ Σ i : Fin n, ∫
    x, condEntExceptCoord (μs := μs) i f x
    ∂μˢ := by
```

**Theorem 3.7** (**v. Gaussian Lipschitz Concentration**, Theorem 5.6 of (Boucheron et al., 2013)). *Let $\boldsymbol{X} = (X_1, \ldots, X_n)$ be a vector of $n$ independent standard Gaussian random variables and $f : \mathbb{R}^n \to \mathbb{R}$ be a $L$-Lipschitz function. Then, for any $t > 0$,*

$$\mathbb{P}\left(|f(\boldsymbol{X}) - \mathbb{E}[f(\boldsymbol{X})]| \geq t\right) \leq 2\exp\left(-\frac{t^2}{2L^2}\right).$$

We formalize Theorem 3.7 as:

```
theorem gaussian_lipschitz_concentration
    {f : (EuclideanSpace ℝ (Fin n)) → ℝ}
    {L : ℝ≥0} (hn : 0 < n) (hL : 0 < L)
    (hf : LipschitzWith L f) (t : ℝ) (ht :
    0 < t) :
  let μ := stdGaussianE n
  (μ {x | t ≤ |f x - ∫ y, f y ∂μ
    |}).toReal ≤ 2 * exp (-t^2 / (2 * (L :
    ℝ)^2)) := by
```

We briefly describe the proof strategy. We formalize the Herbst argument for $f_\epsilon$ defined in Eq. (1). For any $\lambda \in \mathbb{R}$, we apply the Theorem 3.5 to the function $e^{\lambda f_\epsilon(\boldsymbol{X})/2}$

$$\mathrm{Ent}(e^{\lambda f_\epsilon}) \leq 2\mathbb{E}\left\|\nabla e^{\lambda f_\epsilon(\boldsymbol{X})/2}\right\|_2^2 \leq \frac{\lambda^2 L^2}{2}\mathbb{E}\left[e^{\lambda f_\epsilon(\boldsymbol{X})}\right].$$

By differential inequality (we formalize as Gronwall-type ratio bound) and taking limit by Lemma 3.4, we can obtain

$$\log \mathbb{E}\exp\left(\lambda(f(\boldsymbol{X}) - \mathbb{E}f(\boldsymbol{X}))\right) \leq \frac{\lambda^2}{2}L^2,$$

completing the proof of Theorem 3.7 via Chernoff's bound.

### 3.2. Dudley's Entropy Integral Bound

Dudley's bound is the canonical bridge to link covering number with complexity measures. A formal proof of the general Dudley's bound thus supplies a foundational, widely reusable theorem that supports a broad range of theoretical results. Textbook statements (Boucheron et al., 2013; Vershynin, 2018; Wainwright, 2019) typically have incomplete hypotheses such as skipping integrability or hiding the constant. Our formalization uses the following statement.

**Theorem 3.8** (**Dudley's Entropy Integral Bound**). *Let $(A, d)$ be a pseudo-metric space and $s \subseteq A$ a totally bounded set with diameter at most $D > 0$. Let $\{X_t\}_{t \in s}$ be a normalized sub-Gaussian process with parameter $\sigma > 0$, which has integrable exponential-moment for increments and continuous sample paths on $s$, assume that the entropy integral is finite. Then,*

$$\mathbb{E}\left[\sup_{t \in s} X_t\right] \leq 12\sqrt{2}\sigma \cdot \int_0^D \sqrt{\log N(\varepsilon, s, d)}\, d\varepsilon,$$

*where $N(\varepsilon, s)$ denotes the $\varepsilon$-covering number of $s$.*

The formalization of Theorem 3.8 is given by:

```
theorem dudley {μ : Measure Ω}
    [IsProbabilityMeasure μ] {X : A → Ω →
    ℝ} {σ : ℝ} (hσ : 0 < σ) (hX :
    IsSubGaussianProcess μ X σ) {s : Set
    A} (hs : TotallyBounded s) {D : ℝ} (hD
    : 0 < D) (hdiam : Metric.diam s ≤ D)
    (t₀ : A) (ht₀ : t₀ ∈ s) (hcenter : ∀ ω,
    X t₀ ω = 0) (hX_meas : ∀ t, Measurable
    (X t)) (hX_int_exp : ∀ t s : A, ∀ l : ℝ
    , Integrable (fun ω => Real.exp (l *
    (X t ω - X s ω))) μ) (hfinite :
    entropyIntegralENNReal s D ≠ ⊤) (hcont
    : ∀ ω, Continuous (fun (t : ↑s) => X
    t.1 ω)) :
  ∫ ω, ⊔ t ∈ s, X t ω ∂μ ≤ (12 *
    Real.sqrt 2) * σ * entropyIntegral s D
    := by
```

We build `dudley` totally from scratch. We start with $\epsilon$-nets:

```
def IsENet {A : Type*} [PseudoMetricSpace
    A] (t : Finset A) (eps : ℝ) (s : Set
    A) : Prop :=
  s ⊆ ⋃ x ∈ t, closedBall x eps
```

We then define the **covering number** $N(\epsilon, s, d)$ as the minimal cardinality of an $\epsilon$-net:

```
def coveringNumber {A : Type*}
   [PseudoMetricSpace A] (eps : ℝ) (s :
   Set A) : WithTop Nat :=
 sInf {n : WithTop Nat | ∃ t : Finset A,
   IsENet t eps s ∧ (t.card : WithTop
   Nat) = n}
```

We then define the **metric entropy** as the logarithm of the covering number, with appropriate handling of edge cases:

```
def metricEntropy (eps : ℝ) (s : Set A) : ℝ
   :=
 match coveringNumber eps s with
 | ⊤ => 0
 | (n : ℕ) => if n ≤ 1 then 0 else
   Real.log n
```

Taking the square root of entropy, denoted `sqrtEntropy` from `metricEntropy`, we formulate the **entropy integral** via a two-level design. The canonical definition uses extended non-negative reals:

```
def entropyIntegralENNReal (s : Set A) (D
   : ℝ) : ℝ≥0∞ :=
 ∫⁻ eps in Set.Ioc 0 D, ENNReal.ofReal
   (sqrtEntropy eps s)
```

A real-valued wrapper `entropyIntegral` extracts the `toReal` component under finiteness hypothesis.

Next, we formalize **sub-Gaussian processes** via moment generating function bounds, i.e.

```
def IsSubGaussianProcess (μ : Measure Ω)
   (X : A → Ω → ℝ) (σ : ℝ) : Prop :=
 ∀ s t : A, ∀ l : ℝ, μ[fun ω => exp (l *
   (X s ω - X t ω))] ≤
   exp (l^2 * σ^2 * (dist s t)^2 / 2)
```

The **chaining argument** constructs a hierarchy of $\varepsilon$-nets at dyadic scales $\varepsilon_k = D \cdot 2^{-k}$, encapsulated in our `DyadicNets` structure. The key technique is a telescoping decomposition: for any $u \in T_K$, we write $X_u - X_{t_0}$ as a base term from the coarsest net plus increments $\sum_{k=0}^{K-1}(X_{\pi_{k+1}(u)} - X_{\pi_k(u)})$ through successive projections $\pi_k$. Applying finite maximum bounds for sub-Gaussian processes to each increment and summing yields a bound in terms of the dyadic sum $R_K(s, D) := \sum_{k=0}^{K-1} \varepsilon_k \sqrt{\log N(\varepsilon_k, s, d)}$.

The proof then proceeds through two limit arguments. First, we extend from finite nets to a countable dense sequence via Fatou's lemma. Since Fatou requires nonnegative integrands but the supremum may be negative, we introduce a shift function that cancels in expectation. Second, we extend to the uncountable set $s$ by exploiting path continuity which concludes the target. We present a detailed formalization proof in three stages at Section B.

**Implementation challenge:** Lean 4 has two integration formalisms: the nonnegative improper integral $\int^-$ for $\mathbb{R}{\geq}0\infty$ and the interval integral for real-valued functions. $\int^-$ is technically convenient for measure-theoretic arguments such as Fubini above, but it lives in $\mathbb{R}{\geq}0\infty$, so every real-valued bound requires `ofReal`/`toReal` conversions and extra side conditions. Real-valued inequalities are far smoother in $\mathbb{R}$, such as taking limit and integration. Hence we define the entropy integral canonically in `ENNReal`, but state Dudley's bound with `entropyIntegral` for user-friendly downstream use without loss of generality.

# 4. Application: Least Squares Framework

In this section, building on Wainwright (2019, Chapter 13), we present our formalization of the least-squares framework, including linear regression (Section 4.1) and $\ell_1$-constrained regression following Raskutti et al. (2011) (Section 4.2). Both rely on localized capacity control via covering numbers, leveraging the infrastructure developed in Section 3.

To present clearly, we follow Wainwright's approach to focus on the prediction error, which has a direct translation to excess risk via Wainwright (2019, Corollary 14.15).

**Problem Setup.** Consider the nonparametric regression model $y_i = f^*(\boldsymbol{x}_i) + \sigma w_i$ for $i = 1, \ldots, n$, where $w_i \overset{i.i.d.}{\sim} \mathcal{N}(0, 1)$. Given a hypothesis class $\mathcal{F}$, the empirical risk minimizer is $\hat{f} := \operatorname{argmin}_{f \in \mathcal{F}} \frac{1}{n} \sum_{i=1}^n (f(\boldsymbol{x}_i) - y_i)^2$. Our goal is to control the prediction error $\|\hat{f} - f^*\|_n^2 := \frac{1}{n} \sum_{i=1}^n (\hat{f}(\boldsymbol{x}_i) - f^*(\boldsymbol{x}_i))^2$.

We encapsulate this setup in a `RegressionModel` structure and formalize the ERM property:

```
structure RegressionModel (n : ℕ) (X :
   Type*) where
 x : Fin n → X; f_star : X → ℝ; σ : ℝ;
   hσ_pos : 0 < σ
 noiseDistribution : Measure (Fin n → ℝ)
   := stdGaussianPi n

def isLeastSquaresEstimator (y : Fin n → ℝ
   ) (F : Set (X → ℝ)) (x : Fin n → X)
   (f_hat : X → ℝ) : Prop :=
 f_hat ∈ F ∧ ∀ f ∈ F, Σ i, (y i - f_hat
   (x i))^2 ≤ Σ i, (y i - f (x i))^2
```

**Localization.** We define the shifted class $\mathcal{F}^* := \{f - f^* : f \in \mathcal{F}\}$ and assume it is *star-shaped*: $0 \in \mathcal{F}^*$ and $\alpha h \in \mathcal{F}^*$ for all $h \in \mathcal{F}^*$ and $\alpha \in [0, 1]$. The localized Gaussian complexity at radius $\delta$ is

$$\mathcal{G}_n(\mathcal{F}(\delta)) := \mathbb{E}_w \left[ \sup_{\substack{g \in \mathcal{F}^* \\ \|g\|_n \leq \delta}} \left| \frac{1}{n} \sum_{i=1}^n w_i g(\boldsymbol{x}_i) \right| \right]. \quad (4)$$

**Main Results.** The central bridge to link prediction error with Eq. (4) is the critical inequality $\frac{\mathcal{G}_n(\mathcal{F}(\delta))}{\delta} \leq \frac{\delta}{2\sigma}$. Next,

we use Theorem 3.7 to formalize the master error bound (Wainwright, 2019, Theorem 13.5).

**Theorem 4.1** (Master error bound, Theorem 13.5 of (Wainwright, 2019)). *Suppose $\mathcal{F}^*$ is star-shaped, let $\delta_*$ be the smallest positive solution to critical inequality which is the critical radius. Then, for any $t \geq \delta_*$, we have*

$$\mathbb{P}\left(\|f - f^*\|_n^2 \geq 16t\delta_*\right) \leq \exp(-nt\delta_*/2\sigma^2).$$

To obtain $\delta_*$, we need to derive an upper bound on Eq. (4), then the critical inequality in terms of entropy integral becomes solvable. Therefore, we make use of Theorem 3.8 to formalize the following capacity control:

**Theorem 4.2.** *For any star-shaped class $\mathcal{F}^*$, we have*

$$\mathcal{G}_n\left(\mathcal{F}(\delta)\right) \leq \frac{24\sqrt{2}}{\sqrt{n}} \int_0^{2\delta} \sqrt{\log N(\epsilon, \mathcal{F}(\delta), \|\cdot\|_n)}\, d\epsilon.$$

**Remark:** Due to the page limit, we present the Lean details of Theorems 4.1 and 4.2 in Section C.

This reduces the problem to bounding covering numbers, which we demonstrate in two applications to verify the functionality of our framework and shape the formalization standard for covering calculus.

### 4.1. Linear Regression

We consider the linear regression case ($n \geq d$) where the ground truth model is $y_i = \langle \boldsymbol{\theta}, \boldsymbol{x}_i \rangle + \sigma w_i$ associated with the linear predictor class $\mathcal{F} = \{f(\cdot) = \langle \boldsymbol{\theta}, \cdot \rangle : \boldsymbol{\theta} \in \mathbb{R}^d\}$ formalized as `linearPredictorClass`. We apply the general framework to obtain the following rate theorem.

**Theorem 4.3.** *Let $\boldsymbol{X} \in \mathbb{R}^{n \times d}$ be the design matrix, define $r := \mathrm{rank}(\boldsymbol{X})$. Then, for the linear predictor class $\mathcal{F}$,*

$$\mathbb{P}\left(\|\hat{f} - f^*\|_n^2 \leq C_1 \frac{\sigma^2 r}{n}\right) \geq 1 - \exp\left(-C_2 r\right),$$

*for some constants $C_1, C_2 > 0$.*

Our formalization is:

```
theorem linear_minimax_rate_rank (hn : 0 <
   n)
 (M : RegressionModel n (EuclideanSpace ℝ
   (Fin d))) (hf_star : M.f_star ∈
   linearPredictorClass d) (hr : 0 <
   designMatrixRank M.x) (f_hat : (Fin n →
   ℝ) → (EuclideanSpace ℝ (Fin d) → ℝ
   )) (hf_hat : ∀ w,
   isLeastSquaresEstimator (M.response w)
   (linearPredictorClass d) M.x (f_hat
   w)) :
 ∃ C₁ C₂ : ℝ, C₁ > 0 ∧ C₂ > 0 ∧
 (stdGaussianPi n {w | (empiricalNorm n
 (fun i => f_hat w (M.x i) – M.f_star
 (M.x i)))^2 ≤ C₁ * M.σ^2 *
 (designMatrixRank M.x) / n}).toReal ≥
 1 – exp (–C₂ * (designMatrixRank M.x))
 := by
```

where `designMatrixRank` is $r$. We briefly present the formalization strategy. We apply Theorem 4.2 to $\mathcal{F}$ then our goal is to upper bound the covering number. Then, we formalize the following Euclidean reduction

$$\log N(\epsilon, \mathcal{F}(\delta), \|\cdot\|_n) \leq \log N(\epsilon, \mathcal{B}_2^r(\delta), \|\cdot\|_2),$$

where $\mathcal{B}_2^r(\delta)$ is the $\ell_2$ ball of radius $\delta$ on $\mathbb{R}^r$. We then formalize the covering number bound on $\ell_2$ ball (Vershynin, 2018, Corollary 4.2.11).

**Theorem 4.4.** *The covering numbers of $\ell_2$ ball of radius $R$ on $\mathbb{R}^\iota$ satisfy for any $\epsilon > 0$:*

$$N(\epsilon, \mathcal{B}_2^\iota(R), \|\cdot\|_2) \leq \left(1 + \frac{2R}{\epsilon}\right)^\iota.$$

```
theorem coveringNumber_euclideanBall_le
  {R eps : ℝ} (hR : 0 ≤ R)
  (heps : 0 < eps) :
  ((coveringNumber eps (euclideanBall R :
   Set (EuclideanSpace ℝ ι))).untop
   (ne_top_of_lt
   (coveringNumber_lt_top_of_totallyBounded
   heps (euclideanBall_totallyBounded
   R))) : ℝ) ≤ (1 + 2 * R / eps) ^
   Fintype.card ι := by
```

Therefore, we can get $\delta_* = \mathcal{O}(\sqrt{r/n})$ then apply Theorem 4.1 to conclude Theorem 4.3.

### 4.2. High-Dimensional $\ell_1$ Regression

We consider the $\ell_1$-constrained regression (equivalent to Lasso), which allows for $d > n$ case. The function class is $\mathcal{F}_R = \{f(\cdot) = \langle \boldsymbol{\theta}, \cdot \rangle : \boldsymbol{\theta} \in \mathcal{B}_1^d(R)\}$ where $\mathcal{B}_1^d(R)$ is $\ell_1$-ball of radius $R$ on $\mathbb{R}^d$. Following the setting in Raskutti et al. (2011), the key of deriving rate is the Euclidean covering of $\ell_1$-convex hull. We formalize this bound as:

**Lemma 4.5.** *Assume $\boldsymbol{X}$ is normalized column-wise to have $\ell_2$ norm bounded by $\sqrt{n}$. For any $\epsilon > 0$.*

$$N\left(\epsilon, \mathrm{absconv}_1(\boldsymbol{X}/\sqrt{n}; R), \|\cdot\|_2\right) \leq (2d+1)^{\lceil R^2/\epsilon^2 \rceil},$$

*where $\mathrm{absconv}_1(\boldsymbol{X}/\sqrt{n}; R) := \{\boldsymbol{X}\boldsymbol{\theta}/\sqrt{n} : \|\boldsymbol{\theta}\|_1 \leq R\}$.*

The empirical covering follows $\log N(\epsilon, \mathcal{F}_R(\delta), \|\cdot\|_n) \lesssim \frac{R^2}{\epsilon^2} \log d$, which can admit the $\mathcal{O}(R\sqrt{\log d/n})$-rate. The implementation challenge of Lemma 4.5 arises from the need of Maurey's argument, see more details in Section D.

## 5. Human-AI Collaborative Formalization

In this section, we distill three transferable methodological contributions and one critical observation from $\sim 500$ hours of supervised development, supported by $\sim 60$ preserved task specifications. We further classify the declarations in our library by the degree of human intervention required and provide a quantitative breakdown. A self-contained practical recipe is provided in Section E.

## 5.1. Structured Specification Protocol

We empirically find that unstructured instructions without explicit infrastructure pointers fail roughly $70\%$ of the time. We reduce the first-attempt failure rate to roughly $15\%$ via a structured `TASK.md` document with four components:

1. **Target statement.** Besides the exact Lean signature to be proved, we also need to provide the self-contained natural language.

2. **Infrastructure pointers.** Names and file paths of locally available lemmas the proof is expected to invoke[2].

3. **Formalization-oriented proof plan.** A detailed step-by-step plan phrased in tactic-level language rather than purely informal mathematics.

4. **Hard boundaries.** Soft boundaries for agent to follow. See details in Section E.

## 5.2. Iterative Specification Evolution

A single structured `TASK.md` is rarely correct on the first attempt for nontrivial proofs. We consistently observe that *retrying with the same spec essentially never succeeds*; each failure surfaces a distinct ambiguity left implicit by the human, and the productive response is to *refine the spec*. We illustrate this on the density extension of the Gaussian logarithmic Sobolev inequality (GLSI) from $C_c^\infty$ to $W^{1,2}$, which required four spec iterations:

- **v1 (vague strategy).** Hand-waves entropy lower-semicontinuity as "a Fatou-type argument". The agent fails because Mathlib's Fatou demands nonnegative integrands, while $t \log t$ goes negative on $(0, 1)$.

- **v2 (dependencies).** Adds infrastructure pointers to the relevant convergence lemmas but leaves the negativity issue unresolved. persistently reports `sorry`.

- **v3 (decomposition).** Factors out the gradient convergence sub-argument into its own lemma, narrowing the remaining obstacle to entropy convergence alone.

- **v4 (trick, domain, API).** Closes the argument with the shift trick and type domain switch. The agent succeeds on the first attempt.

More details of each specification can be found in Section F. The agent's failures are *informative*: they precisely localize what the human left implicit, rendering the collaboration **convergent** rather than open-ended.

---

[2]This part might not be needed for the current frontier models.

*Table 1.* Percentage of formalized declarations by level of human intervention.

| Level | Description | Percentage |
|---|---|---|
| Autonomous | AI proves alone | $\sim15\%$ |
| Sketch-Guided | AI fills tactic details | $\sim30\%$ |
| Specified | AI proves under full NL proof | $\sim40\%$ |
| Human-critical | Human intervenes | $\sim15\%$ |

## 5.3. Verification of Statement's Consistency

The most dangerous failure mode we encountered is not a failed proof but a *successful proof of a false statement*: a Lean term that type-checks against a misformalized goal. In our library we caught three such cases that had survived multiple rounds of AI's self-judgment: (i) A false "gap" bound requiring control of $N(\varepsilon)/N(2\varepsilon)$, which is *not* implied by `TotallyBounded`; (ii) An impossible disjunction false for specific parameter ranges; (iii) A Riemann rectangle used as an upper bound on a non-monotone integrand.

In each case the agent persistently reported `sorry` and proposed progressively more aggressive structural rewrites; only a human, by constructing explicit counterexamples, detected that the *statement* was wrong. We therefore recommend: (i) human review of every target statement *before* proof attempts, (ii) counterexample checking for any nontrivial inequality, and (iii) treating a persistent `sorry` as a signal to re-verify the *statement*, not merely the proof.

## 5.4. Quantitative Breakdown

By tracing each formalized declaration back to its authoring specification, we partition our library into four levels of human intervention (Table 1). The *Specified* tier dominates: in our experience the bottleneck is rarely tactical proof construction (which the agent handles well) but rather the explicit articulation of a proof that textbooks compress into a single sentence. The *Human-critical* tier captures the non-delegable verification cases above and a small number of theorems whose initial mathematical strategy was flawed and required human-driven re-architecture.

## 6. Conclusion

We present the first large-scale Lean 4 formalization of SLT-approximately 30,000 lines of verified code building all infrastructures *from scratch* through human-AI collaboration. The developed Lean 4 formulation framework includes the high-dimensional Gaussian analysis toolbox and Dudley's entropy integral toolbox, which deepens mathematical understanding and open the door to formulation of modern machine learning theory.

## Acknowledgment

Y. Z. was supported by Warwick Chancellor's International Scholarship. JDL acknowledges support of Open Philanthropy, NSF IIS 2107304, NSF CCF 2212262, ONR Young Investigator Award, NSF CAREER Award 2144994, and NSF CCF 2019844. F. L. was supported by Warwick-SJTU seed fund. We thank Zulip[3] for the project organization tool and Sulis[4] for computation resources.

## Impact Statement

This paper presents work whose goal is to advance the field of Lean 4 for Machine Learning Theory. There might be some potential societal consequences of our work, none which we feel must be specifically highlighted here.

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

## A. List of Key Results

Our formalization library contains more than 1000 theorems/lemmas. In this section, we aim to present the major results of our formalizations and provide an exact reference to locate the statement. The list is shown in Table 2.

*Table 2.* List of our key formalization results with exact reference from textbooks.

| Name | Reference |
|---|---|
| `coveringNumber_lt_top_of_totallyBounded` | Vershynin (2018, Remark 4.2.3) |
| `isENet_of_maximal` | Vershynin (2018, Lemma 4.2.6) |
| `coveringNumber_euclideanBall_le` | Vershynin (2018, Corollary 4.2.13) |
| `coveringNumber_l1Ball_le` | Daras et al. (2021, Theorem 2) |
| `subGaussian_finite_max_bound` | Wainwright (2019, Exercise 2.12) |
| `dudley` | Boucheron et al. (2013, Corollary 13.2) |
| `efronStein` | Boucheron et al. (2013, Theorem 3.1) |
| `gaussianPoincare` | Boucheron et al. (2013, Theorem 3.20) |
| `han_inequality` | Boucheron et al. (2013, Theorem 4.1) |
| `entropy_duality` | Boucheron et al. (2013, Theorem 4.13) |
| `entropy_duality_T` | Boucheron et al. (2013, Remark 4.4) |
| `entropy_subadditive` | Boucheron et al. (2013, Theorem 4.22) |
| `bernoulli_logSobolev` | Boucheron et al. (2013, Theorem 5.1) |
| `gaussian_logSobolev_W12_pi` | Boucheron et al. (2013, Theorem 5.4) |
| `lipschitz_cgf_bound` | Boucheron et al. (2013, Theorem 5.5) |
| `gaussian_lipschitz_concentration` | Boucheron et al. (2013, Theorem 5.6) |
| `local_gaussian_complexity_bound` | Wainwright (2019, (5.48) Gaussian Case) |
| `master_error_bound` | Wainwright (2019, Theorem 13.5) |
| `gaussian_complexity_monotone` | Wainwright (2019, Lemma 13.6) |
| `linear_minimax_rate_rank` | Wainwright (2019, Example 13.8) |
| `bad_event_probability_bound` | Wainwright (2019, Lemma 13.12) |
| `l1BallImage_coveringNumber_le` | Raskutti et al. (2011, Lemma 4, $q = 1$) |

## B. Dudley's Formalization Proof Details

The chaining argument requires constructing a hierarchy of $\varepsilon$-nets at geometrically decreasing scales. We encapsulate this in the `DyadicNets` structure, which provides for each level $k$ a finite set $T_k \subseteq s$ that is an $\varepsilon_k$-net of $s$, where $\varepsilon_k = D \cdot 2^{-k}$ is the dyadic scale at level $k$. For the proof to succeed, we need "good" dyadic nets satisfying the cardinality bound $|T_k| \leq N(\varepsilon_{k+1}, s, d)$. This relates the net size at level $k$ to the covering number at the finer scale $\varepsilon_{k+1} = \varepsilon_k/2$, which is essential for bounding the expected maximum of sub-Gaussian increments at each level later.

Another critical chaining is the dyadic approximation of the entropy integral. Define the dyadic sum

$$R_K(s, D) := \sum_{k=0}^{K-1} \varepsilon_k \cdot \sqrt{\log N(\varepsilon_k, s, d)} \, .$$

This sum approximates the entropy integral via a Riemann-like discretization at geometrically spaced points.

Furthermore, the chaining argument requires defining a sequence of approximations $\pi_0(u), \pi_1(u), \ldots, \pi_K(u) = u$ through the net hierarchy for each point $u$ in the finest net $T_K$. There are two natural approaches:

**Direct projection:** Standard presentations (Boucheron et al., 2013; Vershynin, 2018) define $\pi_k(u)$ as the nearest point in $T_k$ to the original point $u$. The triangle inequality then gives

$$d(\pi_k(u), \pi_{k+1}(u)) \leq d(\pi_k(u), u) + d(u, \pi_{k+1}(u)) \leq \varepsilon_k + \varepsilon_{k+1} = \frac{3}{2}\varepsilon_k \, .$$

**Recursive projection:** An alternative, used in van der vaart & Wellner (2013); Wainwright (2019), defines $\pi_k(u)$ as the nearest point in $T_k$ to $\pi_{k+1}(u)$, i.e. the coarser approximation is chosen to approximate the finer one, not the original point.

For $u \in T_K$:

$$
\pi_k(u) = \begin{cases} u & \text{if } k = K \\ \text{nearest point in } T_k \text{ to } \pi_{k+1}(u) & \text{if } k < K \end{cases}
$$

Our formalization uses the recursive projection, which yields a tighter constant. Since $\pi_{k+1}(u) \in T_{k+1} \subseteq s$ and $T_k$ is an $\varepsilon_k$-net of $s$, we have $d(\pi_k(u), \pi_{k+1}(u)) \leq \varepsilon_k$ directly, without the factor of $3/2$. This improvement from $\frac{3}{2}\varepsilon_k$ to $\varepsilon_k$ propagates through the proof. While modest at each level, it accumulates to a noticeable reduction in the final constant.

Then, the formal proof of `dudley` proceeds in three stages:

## B.1. Stage 1

The first stage establishes a bound for the expected supremum over a finite net. For the net $T_K$ at level $K$, our target is to prove

$$
\mathbb{E}\left[\sup_{u \in T_K} (X_u - X_{t_0})\right] \leq 6\sqrt{2}\sigma \cdot R_{K+1}(s, D) \, .
$$

For any $u \in T_K$, we write

$$
X_u - X_{t_0} = (X_{\pi_0(u)} - X_{t_0}) + \sum_{k=0}^{K-1} (X_{\pi_{k+1}(u)} - X_{\pi_k(u)}) \, ,
$$

where $\pi_k$ denotes the recursive projection to level $k$. The first term is the base term from the coarsest net, and the sum captures the increments as we move through finer nets.

For the *base term*, using the finite maximum bound for sub-Gaussian processes, we have

$$
\mathbb{E}\left[\sup_{u \in T_K} (X_{\pi_0(u)} - X_{t_0})\right] \leq \sigma \epsilon_0 \cdot \sqrt{2 \log |T_0|} \, ,
$$

since all points in the level-0 net are within distance $D$ of $t_0$. The cardinality bound $|T_0| \leq N(\varepsilon_1, s)$ allows us to express this as the first term of $R_{K+1}(s, D)$ scaled by $2\sqrt{2}\sigma$.

For the *increment terms*, the key observations are that these increments have distance at most $\varepsilon_k$ by the recursive projection bound, and that the number of distinct pairs $(\pi_k(u), \pi_{k+1}(u))$ is at most $|T_{k+1}|$ since $\pi_{k+1}(u) \in T_{k+1}$ determines the pair. Applying the finite maximum bound produces a bound of the form:

$$
\mathbb{E}\left[\sup_{u \in T_K} (X_{\pi_{k+1}(u)} - X_{\pi_k(u)})\right] \leq \sigma \varepsilon_k \cdot \sqrt{2 \log |T_{k+1}|} \, .
$$

By the cardinality boundness of "good" nets, we sum over $k$ and re-index to obtain

$$
\mathbb{E}\left[\sup_{u \in T_K} \sum_{k=0}^{K-1} (X_{\pi_{k+1}(u)} - X_{\pi_k(u)})\right]
$$

$$
\leq \sum_{k=0}^{K-1} \sigma \varepsilon_k \cdot \sqrt{2 \log N(\varepsilon_{k+2}, s, d)} \leq 4\sqrt{2}\sigma \cdot R_{K+1}(s, D) \, .
$$

## B.2. Stage 2

This stage extends the bound from finite nets to the countable supremum over a dense sequence $(t_n)_{n \in \mathbb{N}}$ in $s$, where Fatou's lemma enters. For convenience, we work in normalized sub-Gaussian settings ($X_{t_0} \equiv 0$) without loss of generality. The central challenge is that Fatou's lemma requires nonnegative integrands, but the supremum $Y_K := \sup_{u \in T_K} X_u$ may be negative. We resolve this by introducing a shift function

$$
g(\omega) := \inf_{K \in \mathbb{N}} X_{\pi_K(t_0)}(\omega)
$$

where $\pi_K(t_0)$ denotes the projection of $t_0$ onto $T_K$. With shift function in hand, we define $Z_K := Y_K - g \geq 0$. Fatou's lemma gives

$$\mathbb{E}\left[\liminf_{K\to\infty} Z_K\right] \leq \liminf_{K\to\infty} \mathbb{E}[Z_K].$$

By path continuity and sequence density, $\liminf_K Z_K = \sup_{n\in\mathbb{N}} X_{t_n} - g$. Since $\mathbb{E}[Z_K] = \mathbb{E}[Y_K] - \mathbb{E}[g]$, the $\mathbb{E}[g]$ cancels:

$$\mathbb{E}\left[\sup_{n\in\mathbb{N}} X_{t_n}\right] \leq \liminf_{K\to\infty}\left(6\sqrt{2}\sigma \cdot R_{K+1}(s, D)\right).$$

The final step shows

$$\liminf_{K} R_{K+1}(s, D) \leq 2 \cdot \int_0^D \sqrt{\log N(\varepsilon, s, d)}\, d\varepsilon.$$

The approximation $R_K(s, D) \leq 2\cdot(\text{entropy integral}) + 2\cdot(\text{tail})$ holds, where the tail is $\varepsilon_K\sqrt{\log N(\varepsilon_K, s)} \to 0$ as $K \to \infty$. Taking $\liminf$ eliminates the tail, yielding:

$$\mathbb{E}\left[\sup_{n\in\mathbb{N}} X_{t_n}\right] \leq 12\sqrt{2} \cdot \int_0^D \sqrt{\log N(\varepsilon, s, d)}.$$

### B.3. Stage 3

This stage converts the supremum over the uncountable set $s$ to the countable supremum over the dense sequence. Since $s$ is totally bounded in $(A, d)$, it is separable, so there exists a countable dense sequence $(t_n)_{n\in\mathbb{N}} \subseteq s$. By the sample path continuity, for each $\omega$ the map $t \mapsto X_t(\omega)$ is continuous on $s$. Since the supremum of a continuous function over a set equals its supremum over any dense subset, we have

$$\sup_{t\in s} X_t(\omega) = \sup_{n\in\mathbb{N}} X_{t_n}(\omega).$$

Combining this with the bound from the second stage gives

$$\mathbb{E}\left[\sup_{t\in s} X_t\right] \leq 12\sqrt{2}\sigma \cdot \int_0^D \sqrt{\log N(\varepsilon, s, d)}\, d\varepsilon$$

completing the proof.

## C. Lean 4 Formalization of Least Squares

In this section, we present more implementation details of Theorem 4.1 and Theorem 4.2. We aim to show that the formalization exposes and resolves implicit assumptions missed in standard textbooks, enforcing a granular understanding of the theory.

### C.1. Master Error Bound

First, the complete formalization of Theorem 4.1 is:

```
theorem master_error_bound (hn : 0 < n)
  (M : RegressionModel n X) (F : Set (X → ℝ)) (hF_star : M.f_star ∈ F) (δ_star : ℝ) (hδ
    : 0 < δ_star)
  (hCI : satisfiesCriticalInequality n M.σ δ_star (shiftedClass F M.f_star) M.x)
  (hH_star : IsStarShaped (shiftedClass F M.f_star))
  (t : ℝ) (ht : δ_star ≤ t) (f_hat : (Fin n → ℝ) → (X → ℝ))
  (hf_hat : ∀ w, isLeastSquaresEstimator (M.response w) F M.x (f_hat w))
  (hne : (empiricalSphere n (shiftedClass F M.f_star) (Real.sqrt (t * δ_star))
    M.x).Nonempty)
  (hint_u : Integrable (fun w => ⊔ h ∈ localizedBall (shiftedClass F M.f_star) (Real.sqrt
    (t * δ_star)) M.x, |(n : ℝ)⁻¹ * Σ i, w i * h (M.x i)|) (stdGaussianPi n))
  (hint_δ : Integrable (fun w => ⊔ h ∈ localizedBall (shiftedClass F M.f_star) δ_star
    M.x, |(n : ℝ)⁻¹ * Σ i, w i * h (M.x i)|) (stdGaussianPi n))
```

```
(hbdd : ∀ w : Fin n → ℝ, BddAbove {y | ∃ h ∈ localizedBall (shiftedClass F M.f_star)
  (Real.sqrt (t * δ_star)) M.x, y = |(n : ℝ)⁻¹ * Σ i, w i * h (M.x i)|}) :
(stdGaussianPi n {w | (empiricalNorm n (fun i => f_hat w (M.x i) - M.f_star (M.x i)))^2
  ≤ 16 * t * δ_star}).toReal ≥ 1 - exp (-n * t * δ_star / (2 * M.σ^2)) := by
```

Now we will introduce the technical hypotheses:

**hCI:**  The chosen radius $\delta > 0$ should satisfy the critical inequality. This is used to control the probability of the bad event in `bad_event_probability_bound`, which is a standard proof need as Wainwright (2019, Lemma 13.2).

**hH_star:**  The shifted class is star-shaped. The need follows similar reason to `hCI`.

**hne:**  The empirical sphere is non-empty. The bad event is defined in terms of a supremum over the empirical sphere at certain radius. If this sphere is empty, the supremum would be vacuously $-\infty$ or ill-defined. In practice, for most function classes (e.g., linear or constrained class), this holds automatically. But in a formal proof, this must be stated explicitly.

**hint_u:**  Integrability at scale $u$, i.e. the supremum of the empirical process over the localized ball at radius $u = \sqrt{t\delta}$ is integrable with respect to the Gaussian measure. The `bad_event_probability_bound` uses this. The proof involves computing or bounding the expectation of supremum and then applying sub-Gaussian concentration. Both steps require that the supremum random variable is integrable. In the formal proof, Lean's measure theory requires it as an explicit hypothesis.

**hint_δ:**  Integrability at scale $\delta$, same as `hint_u`, but at the radius $\delta$ instead of $u$. The proof bounds the expectation of supremum by relating it to the local Gaussian complexity, which is itself an expectation at scale $\delta$.

**hbdd:**  Boundedness above of the process, which is another technical measure-theoretic condition. The proof takes suprema ($\sqcup$) over localized balls. In Lean 4, $\sqcup$ over an unbounded set can give meaningless results (default to 0 or $\bot$). The BddAbove condition ensures the `iSup` is a genuine supremum.

## C.2. Localized Gaussian Complexity via Dudley

The complete formalization of Theorem 4.2 is:

```
lemma local_gaussian_complexity_bound (n : ℕ) (hn : 0 < n) (H : Set (X → ℝ)) (δ : ℝ) (hδ
  : 0 < δ) (x : Fin n → X) (hH_star : IsStarShaped H)
  (hH_tb : TotallyBounded (empiricalMetricImage n x '' localizedBall H δ x))
  (hfinite : entropyIntegralENNReal (empiricalMetricImage n x '' localizedBall H δ x)
  (2*δ) ≠ ⊤)
  (hcont : ∀ w, Continuous (fun (v : ↑(empiricalMetricImage n x '' localizedBall H δ x)) =
  > innerProductProcess n v.1 w))
  (hint_pos : Integrable (fun w => ⊔ g ∈ localizedBall H δ x, empiricalProcess n x g w)
  (stdGaussianPi n))
  (hint_neg : Integrable (fun w => ⊔ g ∈ localizedBall H δ x, -empiricalProcess n x g w)
  (stdGaussianPi n)) :
  LocalGaussianComplexity n H δ x ≤ (24 * Real.sqrt 2) / Real.sqrt n * entropyIntegral
  (empiricalMetricImage n x '' localizedBall H δ x) (2*δ) := by
```

Now we will introduce the technical hypotheses:

**hfinite:**  The entropy integral is finite. If the entropy integral is infinite, the bound is vacuous. The finiteness in `ENNReal` ensures the real-valued `entropyIntegral` is well-defined and the bound is meaningful. This is a formalization concern: the external `dudley` theorem needs to know the integral converges.

**hcont:**  Pathwise continuity of the process. This is also a formalization need for the external `dudley` theorem. This should hold automatically in finite dimensions, but in a formal proof in Lean, the continuity needs to be stated with respect to the subspace topology on the image set.

**hint_pos:** Integrability of the positive supremum. The proof manipulates integrals (uses linearity of expectation, comparison of integrands), which requires the functions being integrated to be integrable. Lean 4's integral of a non-integrable function is defined to be 0, which would make the bound vacuously useless.

**hint_neg:** Same measure-theoretic regularity as above, but for the negative side of the process. The negative version is needed separately because the supremum of negative function is not simply negative infimum of function in terms of integrability.

### C.3. Observation

The hypotheses `hint_u`, `hint_δ`, `hint_pos`, `hint_neg`, and `hne` are typically missed in standard textbook treatments but act as essential roles to make the statement hold rigorously. The formalization makes precise exactly what must be verified when extending the theory to new settings.

## D. Lean 4 Formalization of Covering of $\ell_1$-Convex Hull

In this section, we present more implementation details of Lemma 4.5. Our formalized statement of Lemma 4.5 is:

```
def l1BallImage (x : Fin n → EuclideanSpace ℝ (Fin d)) (R : ℝ) : Set (EmpiricalSpace n)
  :=
  {v | ∃ θ : EuclideanSpace ℝ (Fin d), l1norm θ ≤ R ∧ v = fun i => (1 / Real.sqrt n) *
  @inner ℝ _ _ θ (x i)}

theorem l1BallImage_coveringNumber_le {R ε : ℝ} (hR : 0 ≤ R) (hε : 0 < ε) (x : Fin n →
    EuclideanSpace ℝ (Fin d)) (hcol : columnNormBound x) (hn : 0 < n) :
  coveringNumber ε (l1BallImage x R) ≤ (2 * d + 1) ^ ⌈R ^ 2 / ε ^ 2⌉₊ := by
```

The major challenge is that the formalization strategy used in Theorem 4.4 can only obtain combinatorial complexity grows exponentially with $d$. So we need to formalize the *probabilistic* rather than combinatorial methods.

In general, our formalization mirrors the Maurey's Empirical Method (Pisier, 2006).

**Step 1: Probabilistic Representation** Any point $v = X\theta/\sqrt{n}$ in $\mathrm{absconv}_1(X/\sqrt{n}; R)$ can be written as an expected value:
$$v = \mathbb{E}[Z]$$
where $Z$ is a random variable taking values in $\{0\} \cup \{\pm R \cdot X_{[:,j]}/\sqrt{n}\}$ with probabilities proportional to $|\theta_j|$.

**Step 2: Variance Control** The random variable $Z$ has bounded second moment:
$$\mathbb{E}[\|Z\|^2] \leq R \cdot \|\theta\|_1$$
This uses the column normalization $\|X_{[:,j]}\|_2 \leq \sqrt{n}$.

**Step 3: Averaging Reduces Variance** For $k$ i.i.d. copies $Z_1, \ldots, Z_k$, the average $\bar{Z}_k = \frac{1}{k}\sum_{\ell=1}^k Z_\ell$ satisfies:
$$\mathbb{E}[\|\bar{Z}_k - v\|^2] \leq \frac{R \cdot \|\theta\|_1}{k} \leq \frac{R^2}{k}$$

**Step 4: Existence via Pigeonhole** Since the expected squared distance is at most $R^2/k$, there exists a specific sample achieving this bound. Setting $k = \lceil R^2/\varepsilon^2 \rceil$ ensures distance $\leq \varepsilon$.

**Step 5: Finite Net Construction** The set of all possible $k$-averages forms a finite net $\mathcal{N}_k$ with:
$$|\mathcal{N}_k| \leq (2d+1)^k$$

since each sample comes from a space of size $2d + 1$.

## E. Practical Recipe for Human-AI Collaborative Formalization

This section complements Section 5 with the practical workflow recipe.

### E.1. Decomposition Strategy

Our central operational principle is to decompose every proof so that each leaf-level lemma can be completed by the agent *within a single context window* without triggering auto-compaction. Empirically, Claude Opus 4.5 handles up to $\sim$300 lines of newly authored Lean per turn before output quality degrades. We thus target $\leq 300$ lines per lemma and prefer many small lemmas over a few large ones. The benefits compound:

- **Higher per-token reasoning budget.** Smaller surface area means the agent spends more of its computation on proof search and less on parsing context.

- **Bounded blast radius.** A failed lemma can be reattempted in isolation without re-instantiating the entire surrounding development.

- **Reusability.** Small lemmas tend to be reusable across the library; large monolithic proofs tend not to be.

- **Reviewability.** Human verification of statements (Section 5.3) is far cheaper at the granularity of small lemmas.

### E.2. Anti-rewrite Instruction

The agent's default response to a large error volume is to abandon the proof. We include the following sentence *verbatim* in every `TASK.md`:

```
DO NOT FREQUENTLY CHANGE THE PROOF. ONLY DO SO WHEN YOU ARE CONFIDENT IT IS WRONG.
FIX ERRORS ONE-BY-ONE.
```

This single instruction was responsible for the largest qualitative improvement we observed during the project.

### E.3. Iterative Cleanup Loop

After a lemma compiles, the resulting code typically contains two classes of artifacts: compiler warnings (unused variables, shadowed binders, deprecated lemmas) and dead `have` statements introduced during exploratory proof search. Both are addressed by dedicated cleanup passes, run in a loop until both report no further changes.

#### E.3.1. WARNING CLEANUP

We use the following prompt:

```
Please modify the formalization code to eliminate the warning messages.
**IMPORTANT:** For unused variables, you need to directly remove them instead of
adding '_', then correct the downstream call of modified results in [LIBRARYNAME].
Please use MCP tools to diagnose.  The target file to modify is [FILENAME]
```

The explicit instruction to remove rather than underscore-prefix is critical: Claude Opus 4.5 strongly prefers the latter, which masks genuine API design issues (e.g. a hypothesis that should never have been added) behind syntactically-clean code.

#### E.3.2. REMOVING UNUSED `have` STATEMENTS

Unused `have` statements accumulate during exploratory proof search and bloat the final development without affecting correctness. We implement the `#check_unused_have` command together with the following operational prompt:

```
Clean up unused have statements from the target Lean file.  First, use the MCP
tool mcp_lean-lsp_lean_file_outline with the absolute file path to get the file
outline and identify all theorem/lemma declarations in the file.  Then, update
@TestUnusedHave.lean to include #check_unused_have command for each declaration
you want to analyze (replace instead of adding).  Run lake build TestUnusedHave
to identify which theorems have unused have statements and note their names.  For
each unused statement reported, use Grep to find its exact location in the file,
then use Edit to remove the entire have statement including any multi-line proof
body.  Work from bottom to top (highest line numbers first) to preserve line
```

```
number accuracy during edits.  For have statements on the same line as other
tactics (like have hX := rfl; linarith), either inline the fact into the tactic
(e.g., linarith [show X from rfl]) or just keep the tactic if it doesn't need the
fact.  After removing all identified statements, run lake build [FILENAME] to
verify compilation succeeds.  Then re-run lake build TestUnusedHave to check for
any cascading unused statements that were exposed (statements that were only used
by the ones you just removed).  Repeat the removal and verification cycle until
all theorems in the file report "No unused have statements found".  Track progress
and provide a summary of total statements removed when complete.  The target Lean
file is [FILENAME].
```

**Run to fixpoint.** Warning cleanup and unused-`have` cleanup interact: removing a dead `have` can expose a new unused-variable warning, and vice versa. We run both passes in alternation until neither produces edits.

## F. Example Task Specifications

### v1: Extension to Strong Gaussian Sobolev Space via Density Argument

Your task is to extend the one-dimensional Gaussian logarithmic Sobolev inequality from twice-differentiable functions with compact support to strong Gaussian Sobolev function class ($f \in W^{1,2}(\gamma)$ with $f \in C^1(\mathbb{R})$).
**STRONG MEANS THE FUNCTION IS CONTINUOUSLY DIFFERENTIABLE**

**Target Theorem Statement.** For any continuously differentiable ($f \in C^1$) function $f : \mathbb{R} \to \mathbb{R}$ with $f \in W^{1,2}(\gamma)$, we have
$$\text{Ent}(f^2) \leq 2\mathbf{E}\left[\|\nabla f(X)\|^2\right].$$

**What We Have Formalized.**

**`FoML/GaussianSobolevDense/Density.lean`.** We have formalized:

- strong Gaussian Sobolev function class (1D specialized version):

```
def MemW12GaussianReal (f : ℝ → ℝ) (γ : Measure ℝ) : Prop :=
  MemLp f 2 γ ∧ MemLp (fun x ↦ fderiv ℝ f x) 2 γ
```

- strong Gaussian Sobolev norm squared

```
noncomputable def GaussianSobolevNormSqReal (f : ℝ → ℝ) (γ : Measure ℝ) : ℝ≥0∞
    :=
  eLpNorm f 2 γ ^ 2 + eLpNorm (fun x ↦ ‖fderiv ℝ f x‖) 2 γ ^ 2
```

- existence of a sequence of smooth compactly supported functions converging to any continuously differentiable $f \in W^{1,2}(\gamma)$ in the 1D Gaussian Sobolev norm

```
theorem exists_smooth_compactSupport_seq_tendsto_real (f : ℝ → ℝ)
    (hf : MemW12GaussianReal f (gaussianReal 0 1))
    (hf_diff : Differentiable ℝ f) (hf_grad_cont : Continuous (fun x => fderiv ℝ
    f x)) :
    ∃ g : ℕ → (ℝ → ℝ),
      (∀ k, ContDiff ℝ (⊤ : ℕ∞) (g k)) ∧
      (∀ k, HasCompactSupport (g k)) ∧
      Tendsto (fun k => GaussianSobolevNormSqReal (f - g k) (gaussianReal 0 1))
    atTop (nhds 0) := by
```

**`FoML/GaussianLSI/OneDimGLSI.lean`.** We have formalized the one-dimensional Gaussian logarithmic Sobolev inequality from twice-differentiable functions with compact support

```
theorem gaussian_logSobolev_CompSmo {f : ℝ → ℝ} (hf : CompactlySupportedSmooth f) :
    LogSobolev.entropy stdGaussianMeasure (fun x => (f x)^2) ≤
    2 * ∫ x, (deriv f x)^2 ∂stdGaussianMeasure := by
```

**Optimal Proof for Formalization.**

**Step 1: Approximating Sequence.**   By the formalized density theorem
`exists_smooth_compactSupport_seq_tendsto_real` in
`FoML/GaussianSobolevDense/Density.lean`, there exists a sequence $\{g_k\}_{k=1}^{\infty} \subset C_c^{\infty}(\mathbb{R}^n)$ such that

$$\|f - g_k\|_{W^{1,2}(\gamma)} \to 0 \quad \text{as } k \to \infty,$$

where $f \in W^{1,2}(\gamma)$ with $f \in C^1(\mathbb{R})$.
This means (by `MemW12GaussianReal` and `GaussianSobolevNormSqReal`):

- $\|f - g_k\|_{L^2(\gamma)} \to 0$, i.e., $\mathbf{E}[(f(X) - g_k(X))^2] \to 0$

- $\|\nabla f - \nabla g_k\|_{L^2(\gamma)} \to 0$, i.e., $\mathbf{E}[\|\nabla f(X) - \nabla g_k(X)\|^2] \to 0$

**Step 2: Convergence of the Gradient Term.**   Since $\nabla g_k \to \nabla f$ in $L^2(\gamma)$, we have

$$\mathbf{E}[\|\nabla g_k(X)\|^2] \to \mathbf{E}[\|\nabla f(X)\|^2] \quad \text{as } k \to \infty.$$

*Proof:* By the triangle inequality in $L^2(\gamma)$,

$$\left| \|\nabla g_k\|_{L^2(\gamma)} - \|\nabla f\|_{L^2(\gamma)} \right| \le \|\nabla g_k - \nabla f\|_{L^2(\gamma)} \to 0.$$

**Step 3: Convergence of $g_k^2$ to $f^2$ in $L^1(\gamma)$.**   We show that $g_k^2 \to f^2$ in $L^1(\gamma)$.
Note that

$$|g_k^2 - f^2| = |g_k - f| \cdot |g_k + f|.$$

By Cauchy–Schwarz:

$$\mathbf{E}[|g_k^2 - f^2|] \le \|g_k - f\|_{L^2(\gamma)} \cdot \|g_k + f\|_{L^2(\gamma)}.$$

Since $g_k \to f$ in $L^2(\gamma)$, the sequence $\{\|g_k\|_{L^2(\gamma)}\}$ is bounded, so $\|g_k + f\|_{L^2(\gamma)}$ is bounded. Therefore,

$$\mathbf{E}[|g_k^2 - f^2|] \to 0.$$

**Step 4: Lower Semicontinuity of Entropy.**   The entropy functional is **lower semicontinuous** with respect to $L^1(\gamma)$ convergence. That is, if $h_k \to h$ in $L^1(\gamma)$ with $h_k, h \ge 0$, then

$$\mathrm{Ent}(h) \le \liminf_{k \to \infty} \mathrm{Ent}(h_k).$$

*Proof of lower semicontinuity:* This follows from the convexity of $\phi(t) = t \log t$ and Fatou-type arguments. Specifically, since $g_k^2 \to f^2$ in $L^1(\gamma)$, by passing to a subsequence we may assume $g_k^2 \to f^2$ almost surely. The function $\phi(t) = t \log t$ (extended to be 0 at $t = 0$) is bounded below, and by a refined version of Fatou's lemma for entropy (or by direct convexity arguments), we obtain the lower semicontinuity.

**Step 5: Completing the Proof.**   By `gaussian_logSobolev_CompSmo` in
`FoML/GaussianLSI/1dGLSI.lean`, for each $g_k \in C_c^{\infty}(\mathbb{R}^n)$, the log-Sobolev inequality holds:

$$\mathrm{Ent}(g_k^2) \le 2\mathbf{E}[\|\nabla g_k(X)\|^2].$$

Taking $\liminf$ as $k \to \infty$:

$$\liminf_{k \to \infty} \mathrm{Ent}(g_k^2) \le \liminf_{k \to \infty} 2\mathbf{E}[\|\nabla g_k(X)\|^2] = 2\mathbf{E}[\|\nabla f(X)\|^2],$$

where the equality uses Step 2.
By lower semicontinuity (Step 4):

$$\mathrm{Ent}(f^2) \le \liminf_{k \to \infty} \mathrm{Ent}(g_k^2).$$

Combining these:

$$\mathrm{Ent}(f^2) \le 2\mathbf{E}[\|\nabla f(X)\|^2].$$

□

**Rules.**

1. Write all formalization code in `FoML/GaussianLSI/OneDimGLSI.lean`.

2. **Success = `lake build` passes + zero sorries + zero custom axioms.** Theorems with sorries/axioms are scaffolding, not results.

3. Make proper use of MCP Tools.

4. DO NOT BE AFRAID OF COMPLEXITY OR LACK OF INFRASTRUCTURES. IF COMPLEX, TRY TO USE A LEMMA-BASED MODULAR APPROACH, DECOMPOSE THE PROOF INTO LEMMAS THEN SOLVE ONE-BY-ONE. IF LACK OF INFRASTRUCTURES, BUILD REQUIRED INFRASTRUCTURES ONE-BY-ONE.

## v2: Lower Semicontinuity of Entropy

**What We Have Formalized.**

**`FoML/GaussianSobolevDense/Density.lean`.** We have formalized:

- strong Gaussian Sobolev function class (1D specialized version):

```
def MemW12GaussianReal (f : ℝ → ℝ) (γ : Measure ℝ) : Prop :=
  MemLp f 2 γ ∧ MemLp (fun x ↦ fderiv ℝ f x) 2 γ
```

- strong Gaussian Sobolev norm squared

```
noncomputable def GaussianSobolevNormSqReal (f : ℝ → ℝ) (γ : Measure ℝ) : ℝ≥0∞
    :=
  eLpNorm f 2 γ ^ 2 + eLpNorm (fun x ↦ ‖fderiv ℝ f x‖) 2 γ ^ 2
```

- existence of a sequence of smooth compactly supported functions converging to any continuously differentiable $f \in W^{1,2}(\gamma)$ in the 1D Gaussian Sobolev norm

```
theorem exists_smooth_compactSupport_seq_tendsto_real (f : ℝ → ℝ)
    (hf : MemW12GaussianReal f (gaussianReal 0 1))
    (hf_diff : Differentiable ℝ f) (hf_grad_cont : Continuous (fun x => fderiv ℝ
    f x)) :
    ∃ g : ℕ → (ℝ → ℝ),
      (∀ k, ContDiff ℝ (⊤ : ℕ∞) (g k)) ∧
      (∀ k, HasCompactSupport (g k)) ∧
      Tendsto (fun k => GaussianSobolevNormSqReal (f - g k) (gaussianReal 0 1))
    atTop (nhds 0) := by
```

**`FoML/GaussianLSI/Entropy.lean`.** We have formalized:

- definition of entropy

```
def entropy (μ : Measure Ω) (f : Ω → ℝ) : ℝ :=
  ∫ ω, f ω * log (f ω) ∂μ - (∫ ω, f ω ∂μ) * log (∫ ω, f ω ∂μ)
```

- entropy of $f^2$ with respect to the normalization

```
def entropySquare (μ : Measure Ω) (f : Ω → ℝ) : ℝ :=
  entropy μ (fun ω => (f ω)^2)
```

- properties of entropy (please read carefully and use properly):

  1. `entropy_const`: Entropy of a constant function is zero.
  2. `entropy_congr`: If $f = g$ a.e., then their entropies are equal.
  3. `jensen_mul_log`: Jensen's inequality for the convex function $x \cdot \log(x)$.

4. `entropy_nonneg`: Entropy is nonnegative for probability measures and nonnegative integrands.

5. `entropySquare_eq`: The entropy of $f^2$ is always well-defined in the sense that the integrand $f^2 \cdot \log(f^2) = 2 \cdot f^2 \cdot \log|f|$ is measurable when $f$ is.

6. `entropy_sq_normalized`, `log_sq_eq_two_mul_log_abs`.

7. `entropy_sq_abs_log`: Entropy of $f^2$ in terms of $2 \cdot f^2 \cdot \log|f|$.

**`FoML/FatouReal.lean`.** **Fatou's Lemma for Nonnegative Real Functions**: For a sequence of nonnegative measurable integrable functions with integrable liminf:

```
theorem integral_liminf_le_of_nonneg
    {f : ℕ → Ω → ℝ}
    (hf_meas : ∀ n, Measurable (f n))
    (hf_nonneg : ∀ n ω, 0 ≤ f n ω)
    (hf_int : ∀ n, Integrable (f n) μ)
    (h_liminf_int : Integrable (fun ω => liminf (fun n => f n ω) atTop) μ)
    (h_cobdd : IsCoboundedUnder (· ≥ ·) atTop (fun n => ∫ ω, f n ω ∂μ)) :
    ∫ ω, liminf (fun n => f n ω) atTop ∂μ ≤
    liminf (fun n => ∫ ω, f n ω ∂μ) atTop := by
```

This is derived from the `ENNReal` Fatou lemma via the transformation:

1. Embed via `ENNReal.ofReal`.

2. Apply `lintegral_liminf_le`.

3. Convert back using integral–lintegral relationships.

Note: The `h_cobdd` assumption ensures the real-valued liminf is well-defined (not the junk value 0 from an unbounded sequence). This holds automatically when integrals are bounded above.

**`FoML/ConvergenceL1Subseq.lean`.** Convergence in $L^1(\mu)$ yields an a.e.-convergent subsequence

```
theorem exists_seq_tendsto_ae_of_tendsto_eLpNorm_one
    [NormedAddCommGroup E] {f : ℕ → α → E} {g : α → E}
    (hf : ∀ n, AEStronglyMeasurable (f n) mu)
    (hg : AEStronglyMeasurable g mu)
    (hfg : Tendsto (fun n => eLpNorm (f n - g) (1 : ENNReal) mu) atTop (nhds 0)) :
    ∃ ns : ℕ → ℕ, StrictMono ns ∧
      ∀ᵐ x ∂mu, Tendsto (fun i => f (ns i) x) atTop (nhds (g x)) := by
```

**Target Theorem Statement.** By `exists_smooth_compactSupport_seq_tendsto_real`, we can construct a sequence of smooth compactly supported functions $\{g_k\}_{k=1}^{\infty} \subset C_c^{\infty}(\mathbb{R})$ converging to any $f$ in the strong Gaussian Sobolev function class ($f \in W^{1,2}(\gamma)$ with $f \in C^1(\mathbb{R})$) in the 1D Gaussian Sobolev norm.
**STRONG MEANS THE FUNCTION IS CONTINUOUSLY DIFFERENTIABLE**
Therefore, our target theorem is a consequent result:

$$\mathrm{Ent}(f^2) \leq \liminf_{k \to \infty} \mathrm{Ent}(g_k^2).$$

**Optimal Proof for Formalization.**

**Step 1: Approximating Sequence.** By the formalized density theorem `exists_smooth_compactSupport_seq_tendsto_real` in `FoML/GaussianSobolevDense/Density.lean`, there exists a sequence $\{g_k\}_{k=1}^{\infty} \subset C_c^{\infty}(\mathbb{R})$ such that

$$\|f - g_k\|_{W^{1,2}(\gamma)} \to 0 \quad \text{as } k \to \infty,$$

where $f \in W^{1,2}(\gamma)$ with $f \in C^1(\mathbb{R})$.
By `MemW12GaussianReal` and `GaussianSobolevNormSqReal`, we can obtain:

$$\|f - g_k\|_{L^2(\gamma)} \to 0, \quad \text{i.e., } \mathbf{E}[(f(X) - g_k(X))^2] \to 0.$$

**Step 2: Convergence of $g_k^2$ to $f^2$ in $L^1(\gamma)$.** We show that $g_k^2 \to f^2$ in $L^1(\gamma)$. Note that

$$|g_k^2 - f^2| = |g_k - f| \cdot |g_k + f|.$$

By Cauchy-Schwarz (Use `LeanSearch` MCP Tool to search for the Holder's inequality and use proper conjugate pair $p$ and $q$):

$$\mathbf{E}[|g_k^2 - f^2|] \le \|g_k - f\|_{L^2(\gamma)} \cdot \|g_k + f\|_{L^2(\gamma)}.$$

Since $g_k \to f$ in $L^2(\gamma)$, the sequence $\{\|g_k\|_{L^2(\gamma)}\}$ is bounded, so $\|g_k + f\|_{L^2(\gamma)}$ is bounded. Therefore,

$$\mathbf{E}[|g_k^2 - f^2|] \to 0.$$

**Step 3: Lower Semicontinuity of Entropy.** The entropy functional is **lower semicontinuous** with respect to $L^1(\gamma)$ convergence. That is, if $h_k \to h$ in $L^1(\gamma)$ with $h_k, h \ge 0$, then

$$\mathrm{Ent}(h) \le \liminf_{k \to \infty} \mathrm{Ent}(h_k).$$

1. Take $h_k = g_k^2$ and $h = f^2$.

2. Proof needs a series of integrability, **remember all integrabilities can be proven by the compactness of** $\{g_k\}_{k=1}^{\infty} \subset C_c^{\infty}(\mathbb{R}^n)$ **and $f$ belongs to strong Gaussian Sobolev space ($f \in L^2(\gamma)$ and $f' \in L^2(\gamma)$).**

3. Following from the convexity of $\phi(t) = t \log t$ and real-valued Fatou lemma `integral_liminf_le_of_nonneg`. Specifically, since $g_k^2 \to f^2$ in $L^1(\gamma)$, by `exists_seq_tendsto_ae_of_tendsto_eLpNorm_one`, there exists a subsequence $g_{k'}^2 \to f^2$ almost surely. The function $\phi(t) = t \log t$ (extended to be 0 at $t = 0$) is bounded below, and by `integral_liminf_le_of_nonneg` for entropy, we obtain the lower semicontinuity.

**Rules.**

1. Create a file in `FoML/GaussianSobolevDense/` named `LiminfContEnt.lean` and write all formalization code in this file.

2. **Success = `lake build` passes + zero sorries + zero custom axioms.** Theorems with sorries/axioms are scaffolding, not results.

3. Make proper use of MCP Tools.

4. DO NOT BE AFRAID OF COMPLEXITY OR LACK OF INFRASTRUCTURES. IF COMPLEX, TRY TO USE A LEMMA-BASED MODULAR APPROACH, DECOMPOSE THE PROOF INTO LEMMAS THEN SOLVE ONE-BY-ONE. IF LACK OF INFRASTRUCTURES, BUILD REQUIRED INFRASTRUCTURES ONE-BY-ONE.

## v3: Convergence of the Gradient Term in $L^2(\gamma)$

**What We Have Formalized.**
**`FoML/GaussianSobolevDense/Density.lean`.** We have formalized:

- strong Gaussian Sobolev function class (1D specialized version):

```
def MemW12GaussianReal (f : ℝ → ℝ) (γ : Measure ℝ) : Prop :=
  MemLp f 2 γ ∧ MemLp (fun x ↦ fderiv ℝ f x) 2 γ
```

- strong Gaussian Sobolev norm squared

```
noncomputable def GaussianSobolevNormSqReal (f : ℝ → ℝ) (γ : Measure ℝ) : ℝ≥0∞
    :=
  eLpNorm f 2 γ ^ 2 + eLpNorm (fun x ↦ ‖fderiv ℝ f x‖) 2 γ ^ 2
```

- existence of a sequence of smooth compactly supported functions converging to any continuously differentiable $f \in W^{1,2}(\gamma)$ in the 1D Gaussian Sobolev norm

```
theorem exists_smooth_compactSupport_seq_tendsto_real (f : ℝ → ℝ)
    (hf : MemW12GaussianReal f (gaussianReal 0 1))
    (hf_diff : Differentiable ℝ f) (hf_grad_cont : Continuous (fun x => fderiv ℝ
    f x)) :
    ∃ g : ℕ → (ℝ → ℝ),
      (∀ k, ContDiff ℝ (⊤ : ℕ∞) (g k)) ∧
      (∀ k, HasCompactSupport (g k)) ∧
      Tendsto (fun k => GaussianSobolevNormSqReal (f - g k) (gaussianReal 0 1))
    atTop (nhds 0) := by
```

**FoML/GaussianLSI/Entropy.lean.**  We have formalized:

- definition of entropy

```
def entropy (μ : Measure Ω) (f : Ω → ℝ) : ℝ :=
  ∫ ω, f ω * log (f ω) ∂μ - (∫ ω, f ω ∂μ) * log (∫ ω, f ω ∂μ)
```

- entropy of $f^2$ with respect to the normalization

```
def entropySquare (μ : Measure Ω) (f : Ω → ℝ) : ℝ :=
  entropy μ (fun ω => (f ω)^2)
```

- properties of entropy (please read carefully and use properly):

  1. entropy_const: Entropy of a constant function is zero.
  2. entropy_congr: If $f = g$ a.e., then their entropies are equal.
  3. jensen_mul_log: Jensen's inequality for the convex function $x \cdot \log(x)$.
  4. entropy_nonneg: Entropy is nonnegative for probability measures and nonnegative integrands.
  5. entropySquare_eq: The entropy of $f^2$ is always well-defined in the sense that the integrand $f^2 \cdot \log(f^2) = 2 \cdot f^2 \cdot \log|f|$ is measurable when $f$ is.
  6. entropy_sq_normalized, log_sq_eq_two_mul_log_abs.
  7. entropy_sq_abs_log: Entropy of $f^2$ in terms of $2 \cdot f^2 \cdot \log|f|$.

**FoML/GaussianSobolevDense/LiminfContEnt.lean.**  We have formalized:

- convergence in Gaussian Sobolev norm (squared) implies convergence in $L^2(\gamma)$:

```
lemma tendsto_eLpNorm_sub_of_sobolev (f : ℝ → ℝ) (g : ℕ → ℝ → ℝ) (μ : Measure ℝ
    )
    (h_tend : Tendsto (fun k => GaussianSobolevNormSqReal (f - g k) μ) atTop
    (nhds 0)) :
    Tendsto (fun k => eLpNorm (f - g k) 2 μ) atTop (nhds 0) := by
```

- convergence in $L^2(\gamma)$ of real-valued functions implies convergence in $L^1(\gamma)$ of their pointwise squares:

```
lemma tendsto_eLpNorm_sub_of_sobolev (f : ℝ → ℝ) (g : ℕ → ℝ → ℝ) (μ : Measure ℝ
    )
    (h_tend : Tendsto (fun k => GaussianSobolevNormSqReal (f - g k) μ) atTop
    (nhds 0)) :
    Tendsto (fun k => eLpNorm (f - g k) 2 μ) atTop (nhds 0) := by
```

**Target Theorem Statement.**  By exists_smooth_compactSupport_seq_tendsto_real, we can construct a sequence of smooth compactly supported functions $\{g_k\}_{k=1}^{\infty} \subset C_c^{\infty}(\mathbb{R})$ converging to any $f$ in the strong Gaussian Sobolev function class ($f \in W^{1,2}(\gamma)$ with $f \in C^1(\mathbb{R})$) in the 1D Gaussian Sobolev norm.
**STRONG MEANS THE FUNCTION IS CONTINUOUSLY DIFFERENTIABLE**
Therefore, our target formalizations (as a lemma) are:

$$\int \|\nabla g_k\|_2^2 \, d\gamma \longrightarrow \int \|\nabla f\|_2^2 \, d\gamma.$$

**Optimal Proof for Formalization.**

**Step 1: Approximating Sequence.** By the formalized density theorem
`exists_smooth_compactSupport_seq_tendsto_real` in
`FoML/GaussianSobolevDense/Density.lean`, there exists a sequence $\{g_k\}_{k=1}^{\infty} \subset C_c^{\infty}(\mathbb{R})$ such that

$$\|f - g_k\|_{W^{1,2}(\gamma)} \to 0 \quad \text{as } k \to \infty,$$

where $f \in W^{1,2}(\gamma)$ with $f \in C^1(\mathbb{R})$.
By `MemW12GaussianReal` and `GaussianSobolevNormSqReal`, we can obtain:

$$\|\nabla f - \nabla g_k\|_{L^2(\gamma)} \to 0, \quad \text{i.e.,} \quad \mathbf{E}[\|\nabla f(X) - \nabla g_k(X)\|^2] \to 0.$$

**Step 2: Convergence of the Gradient Term.** Since $\nabla g_k \to \nabla f$ in $L^2(\gamma)$, we have

$$\mathbf{E}[\|\nabla g_k(X)\|^2] \to \mathbf{E}[\|\nabla f(X)\|^2] \quad \text{as } k \to \infty.$$

*Proof sketch:* By the triangle inequality in $L^2(\gamma)$,

$$\left| \|\nabla g_k\|_{L^2(\gamma)} - \|\nabla f\|_{L^2(\gamma)} \right| \le \|\nabla g_k - \nabla f\|_{L^2(\gamma)} \to 0.$$

**Rules.**

1. Write all formalization code in `LiminfContEnt.lean`.

2. **Success = `lake build` passes + zero sorries + zero custom axioms.** Theorems with sorries/axioms are scaffolding, not results.

3. Make proper use of MCP Tools.

4. DO NOT BE AFRAID OF COMPLEXITY OR LACK OF INFRASTRUCTURES. IF COMPLEX, TRY TO USE A LEMMA-BASED MODULAR APPROACH, DECOMPOSE THE PROOF INTO LEMMAS THEN SOLVE ONE-BY-ONE. IF LACK OF INFRASTRUCTURES, BUILD REQUIRED INFRASTRUCTURES ONE-BY-ONE.

## v4: GLSI Extension to Strong Gaussian Sobolev Space

**What We Have Formalized.**
**`FoML/GaussianSobolevDense/Density.lean`.** We have formalized:
- strong Gaussian Sobolev function class (1D specialized version):

```
def MemW12GaussianReal (f : ℝ → ℝ) (γ : Measure ℝ) : Prop :=
  MemLp f 2 γ ∧ MemLp (fun x ↦ fderiv ℝ f x) 2 γ
```

**NOTICE THAT YOU CAN PROVE MOST INTEGRABILITY OR FINITENESS FROM `MemW12GaussianReal`.**

- strong Gaussian Sobolev norm squared

```
noncomputable def GaussianSobolevNormSqReal (f : ℝ → ℝ) (γ : Measure ℝ) : ℝ≥0∞
    :=
  eLpNorm f 2 γ ^ 2 + eLpNorm (fun x ↦ ‖fderiv ℝ f x‖) 2 γ ^ 2
```

- existence of a sequence of smooth compactly supported functions converging to any continuously differentiable $f \in W^{1,2}(\gamma)$ in the 1D Gaussian Sobolev norm

```
theorem exists_smooth_compactSupport_seq_tendsto_real (f : ℝ → ℝ)
    (hf : MemW12GaussianReal f (gaussianReal 0 1))
```

```
      (hf_diff : Differentiable ℝ f) (hf_grad_cont : Continuous (fun x => fderiv ℝ
   f x)) :
   ∃ g : ℕ → (ℝ → ℝ),
      (∀ k, ContDiff ℝ (⊤ : ℕ∞) (g k)) ∧
      (∀ k, HasCompactSupport (g k)) ∧
      Tendsto (fun k => GaussianSobolevNormSqReal (f - g k) (gaussianReal 0 1))
   atTop (nhds 0) := by
```

**FoML/GaussianLSI/Entropy.lean.**  We have formalized:

- definition of entropy

```
def entropy (μ : Measure Ω) (f : Ω → ℝ) : ℝ :=
  ∫ ω, f ω * log (f ω) ∂μ - (∫ ω, f ω ∂μ) * log (∫ ω, f ω ∂μ)
```

- entropy of $f^2$ with respect to the normalization

```
def entropySquare (μ : Measure Ω) (f : Ω → ℝ) : ℝ :=
  entropy μ (fun ω => (f ω)^2)
```

- properties of entropy (please read carefully and use properly):

  1. `entropy_const`: Entropy of a constant function is zero.
  2. `entropy_congr`: If $f = g$ a.e., then their entropies are equal.
  3. `jensen_mul_log`: Jensen's inequality for the convex function $x \cdot \log(x)$.
  4. `entropy_nonneg`: Entropy is nonnegative for probability measures and nonnegative integrands.
  5. `entropySquare_eq`: The entropy of $f^2$ is always well-defined in the sense that the integrand $f^2 \cdot \log(f^2) = 2 \cdot f^2 \cdot \log|f|$ is measurable when $f$ is.
  6. `entropy_sq_normalized`, `log_sq_eq_two_mul_log_abs`.
  7. `entropy_sq_abs_log`: Entropy of $f^2$ in terms of $2 \cdot f^2 \cdot \log|f|$.

**FoML/GaussianSobolevDense/LiminfContEnt.lean.**  We have formalized:

- $\phi(t) = t \cdot \log(t)$ is bounded below by $-1/e$ for $t \geq 0$:

```
lemma mul_log_ge_neg_inv_exp (t : ℝ) (ht : 0 ≤ t) : -(1 / Real.exp 1) ≤ t *
    Real.log t := by
```

- convergence in Gaussian Sobolev norm (squared) implies convergence in $L^2(\gamma)$:

```
lemma tendsto_eLpNorm_sub_of_sobolev (f : ℝ → ℝ) (g : ℕ → ℝ → ℝ) (μ : Measure ℝ
    )
    (h_tend : Tendsto (fun k => GaussianSobolevNormSqReal (f - g k) μ) atTop
    (nhds 0)) :
    Tendsto (fun k => eLpNorm (f - g k) 2 μ) atTop (nhds 0) := by
```

- convergence in $L^2(\gamma)$ of real-valued functions implies convergence in $L^1(\gamma)$ of their pointwise squares:

```
lemma tendsto_eLpNorm_sub_of_sobolev (f : ℝ → ℝ) (g : ℕ → ℝ → ℝ) (μ : Measure ℝ
    )
    (h_tend : Tendsto (fun k => GaussianSobolevNormSqReal (f - g k) μ) atTop
    (nhds 0)) :
    Tendsto (fun k => eLpNorm (f - g k) 2 μ) atTop (nhds 0) := by
```

- convergence in Gaussian Sobolev norm (squared) implies convergence of gradient in $L^2(\gamma)$:

```
lemma tendsto_integral_norm_fderiv_sq_of_sobolev (f : ℝ → ℝ) (g : ℕ → ℝ → ℝ)
    (μ : Measure ℝ)
    (hf_diff : Differentiable ℝ f) (hf_grad_cont : Continuous (fun x => fderiv ℝ
    f x))
    (hg_diff : ∀ k, Differentiable ℝ (g k))
    (hg_grad_cont : ∀ k, Continuous (fun x => fderiv ℝ (g k) x))
    (hf_mem : MemLp (fun x => ‖fderiv ℝ f x‖) 2 μ)
    (hg_mem : ∀ k, MemLp (fun x => ‖fderiv ℝ (g k) x‖) 2 μ)
    (h_tend : Tendsto (fun k => GaussianSobolevNormSqReal (f - g k) μ) atTop
    (nhds 0)) :
    Tendsto (fun k => ∫ x, ‖fderiv ℝ (g k) x‖^2 ∂μ) atTop
      (nhds (∫ x, ‖fderiv ℝ f x‖^2 ∂μ)) := by
```

**FoML/GaussianLSI/OneDimGLSI.lean.**  We have formalized the one-dimensional Gaussian logarithmic
Sobolev inequality for twice-differentiable functions with compact support

```
theorem gaussian_logSobolev_CompSmo_fderiv {f : ℝ → ℝ} (hf :
    CompactlySupportedSmooth f) :
    LogSobolev.entropy stdGaussianMeasure (fun x => (f x)^2) ≤
    2 * ∫ x, ‖fderiv ℝ f x‖^2 ∂stdGaussianMeasure := by
```

**FoML/ConvergenceL1Subseq.lean.**  Convergence in $L^1(\mu)$ yields an a.e.-convergent subsequence

```
theorem exists_seq_tendsto_ae_of_tendsto_eLpNorm_one
    [NormedAddCommGroup E] {f : ℕ → α → E} {g : α → E}
    (hf : ∀ n, AEStronglyMeasurable (f n) mu)
    (hg : AEStronglyMeasurable g mu)
    (hfg : Tendsto (fun n => eLpNorm (f n - g) (1 : ENNReal) mu) atTop (nhds 0)) :
    ∃ ns : ℕ → ℕ, StrictMono ns ∧
      ∀ᵐ x ∂mu, Tendsto (fun i => f (ns i) x) atTop (nhds (g x)) := by
```

**Target Theorem Statement.**  By `exists_smooth_compactSupport_seq_tendsto_real`, we can construct
a sequence of smooth compactly supported functions $\{g_k\}_{k=1}^{\infty} \subset C_c^{\infty}(\mathbb{R})$ converging to any $f$ in the strong Gaussian
Sobolev function class ($f \in W^{1,2}(\gamma)$ with $f \in C^1(\mathbb{R})$) in the 1D Gaussian Sobolev norm.
Your task is to extend the one-dimensional Gaussian logarithmic Sobolev inequality from twice-differentiable functions
with compact support to strong Gaussian Sobolev function class ($f \in W^{1,2}(\gamma)$ with $f \in C^1(\mathbb{R})$).
**STRONG MEANS THE FUNCTION IS CONTINUOUSLY DIFFERENTIABLE**
Therefore, our **TARGET THEOREM** is:

For any continuously differentiable ($f \in C^1$) function $f : \mathbb{R} \to \mathbb{R}$ with $f \in W^{1,2}(\gamma)$, we have

$$\text{Ent}(f^2) \leq 2\mathbf{E}\left[\|\nabla f(X)\|^2\right].$$

**Optimal Proof for Formalization.**

**Step 1: Preliminaries about** $(x \log x)$**, WELL-FORMALIZED.**  Define $\phi : [0, \infty) \to \mathbb{R}$ by $\phi(x) = x \log x$ (has
been formalized as `def phi`).
**Fact 1 (continuity).** $\phi$ is continuous on $[0, \infty)$. This has been formalized as `phi_continuous`.
**Fact 2 (bounded below).**

$$\inf_{x \geq 0} \phi(x) = -\frac{1}{e}.$$

Hence

$$\phi(x) + \frac{1}{e} \geq 0 \quad \text{for all } x \geq 0. \tag{1}$$

This "shift to nonnegativity" is what makes Fatou immediately applicable, which has been formalized in
`mul_log_ge_neg_inv_exp`.

**Step 2: A subsequence lemma (from $L^1$ to a.e.), WELL-FORMALIZED. Lemma (a.e. convergent subsequence).** Convergence in $L^1(\mu)$ yields an a.e.-convergent subsequence, has been formalized as `exists_seq_tendsto_ae_of_tendsto_eLpNorm_one` in `FoML/ConvergenceL1Subseq.lean`.

**Step 3: Lower semicontinuity of $\int g \log g$ via Fatou's lemma.** IMPORTANT: FROM HERE, WE WORK IN ENNREAL!!!

**Proposition (lower semicontinuity).** Assume $q_k \geq 0$ and $q_k \to q$ in $L^1(\gamma)$. Then

$$\int \phi(q) \, d\gamma \leq \liminf_{k \to \infty} \int \phi(q_k) \, d\gamma, \quad \text{i.e.} \quad \int q \log q \, d\gamma \leq \liminf_{k \to \infty} \int q_k \log q_k \, d\gamma, \tag{3}$$

**allowing the value $+\infty$ on either side.** (We will set $q_k = g_k^2$, $q = f^2$ later.)

*Proof.* Let

$$L := \liminf_{k \to \infty} \int \phi(q_k) \, d\gamma \in [-1/e, \infty].$$

Choose a subsequence (still denoted $q_k$ for simplicity) such that

$$\int \phi(q_k) \, d\gamma \to L. \tag{4}$$

(4) can be proven via: set

$$a_k := \int \phi(q_k) \, d\mu \in \left[-\frac{1}{e}, \infty\right] \quad \text{where } \phi(x) = x \log x.$$

Let

$$L := \liminf_{k \to \infty} a_k \in \left[-\frac{1}{e}, \infty\right].$$

If $L < \infty$, then there exists a subsequence $(a_{k_j}^*)_{j \geq 1}$ such that

$$a_{k_j}^* \to L.$$

If $L = \infty$, then there is nothing to choose: "$\leq \liminf$" is automatic whenever the right-hand side is $+\infty$.
Define the tail infima

$$b_j := \inf_{k \geq j} a_k.$$

Then $(b_j)$ is nondecreasing and $b_j \uparrow L$ by definition of $\liminf$.
For each $j$, by the definition of infimum, there exists some index $k_j \geq j$ such that

$$a_{k_j} \leq b_j + \frac{1}{j}.$$

Then,

$$b_j \leq a_{k_j} \leq b_j + \frac{1}{j},$$

so as $j \to \infty$,

$$a_{k_j} \to L.$$

This is the subsequence you used. It is a standard "almost minimizer of tail infimum" construction.
Given $q_k \to q$ in $L^1(\gamma)$, apply the lemma `exists_seq_tendsto_ae_of_tendsto_eLpNorm_one` to extract a **subsequence** such that $q_k \to q$ a.e.
By continuity of $\phi$ (Fact 1), we have $\phi(q_k) \to \phi(q)$ a.e., hence

$$\phi(q) \leq \liminf_{k \to \infty} \phi(q_k) \quad \text{a.e.} \tag{5}$$

Now shift to nonnegativity using (1): define

$$h_k := \phi(q_k) + \frac{1}{e} \geq 0, \qquad h := \phi(q) + \frac{1}{e} \geq 0.$$

From (5),

$$h \leq \liminf_k h_k \quad \text{a.e.}$$

Fatou's lemma for nonnegative functions (`lintegral_liminf_le!!!`) gives

$$\int h \, d\gamma \leq \liminf_{k \to \infty} \int h_k \, d\gamma.$$

Subtract $1/e$ from both sides (note $\gamma$ is a probability measure, so $\int \frac{1}{e} \, d\gamma = \frac{1}{e}$) to get

$$\int \phi(q) \, d\gamma \leq \liminf_{k \to \infty} \int \phi(q_k) \, d\gamma.$$

Using (4), the right-hand side equals $L$, which is $\liminf_k \int \phi(g_k)$. This proves (3). ∎

**Integrability remarks.**

- We did **not** assume $\phi(g_k) \in L^1$. If $\int \phi(g_k) = +\infty$ along a subsequence, Fatou still applies because $h_k \geq 0$ and the integrals are allowed to be $+\infty$.

- The negative part is never a problem because $\phi \geq -1/e$; that is exactly why the shift works.

**Step 4: Lower semicontinuity of entropy.** Define, for $q \geq 0$,

$$\text{Ent}(q) = \int q \log q \, d\gamma - m \log m, \qquad m := \int q \, d\gamma.$$

**Corollary.** If $q_k \geq 0$ and $q_k \to q$ in $L^1(\gamma)$, then

$$\text{Ent}(q) \leq \liminf_{k \to \infty} \text{Ent}(q_k). \tag{6}$$

*Proof.* From $L^1$-convergence, $m_k := \int q_k \, d\gamma \to \int q \, d\gamma = m$. The map $x \mapsto x \log x$ is continuous on $[0, \infty)$, so

$$m_k \log m_k \to m \log m. \tag{7}$$

From the proposition,

$$\int q \log q \, d\gamma \leq \liminf_{k \to \infty} \int q_k \log q_k \, d\gamma. \tag{8}$$

Combine (7)–(8):

$$\text{Ent}(q) = \int q \log q \, d\gamma - m \log m \leq \liminf_k \int q_k \log q_k \, d\gamma - \lim_k m_k \log m_k = \liminf_k \text{Ent}(q_k).$$

∎

**Step 5: Application in Our Case.** Take $q_k = f_k^2$ and $q = f^2$. So the corollary gives

$$\text{Ent}(f^2) \leq \liminf_{k \to \infty} \text{Ent}(g_k^2),$$

which is the exact lower semicontinuity.

**Step 6: Back to Real Space.** By `tendsto_integral_norm_fderiv_sq_of_sobolev` and `gaussian_logSobolev_CompSmo_fderiv`, we can obtain

$$\liminf_{k \to \infty} \text{Ent}(g_k) \leq \liminf_{k \to \infty} 2 \int |\nabla g_k|^2 \, d\gamma = 2 \int |\nabla f|^2 \, d\gamma < \infty. \tag{$*$}$$

Now combine $(*)$ with

$$\text{Ent}(f^2) \leq \liminf_{k \to \infty} \text{Ent}(g_k^2),$$

to get all finiteness, then we can convert back to real space.

**Conclude the Limit.**
$$\mathrm{Ent}(f^2) \leq 2\mathbf{E}\left[\|\nabla f(X)\|^2\right].$$

follows from results formalized, add the proof by yourself.

**Rules.**

1. Write all formalization code in `LiminfContEnt.lean`.

2. **Success = `lake build` passes + zero sorries + zero custom axioms.** Theorems with sorries/axioms are scaffolding, not results.

3. Make proper use of MCP Tools.

4. DO NOT BE AFRAID OF COMPLEXITY OR LACK OF INFRASTRUCTURES. IF COMPLEX, TRY TO USE A LEMMA-BASED MODULAR APPROACH, DECOMPOSE THE PROOF INTO LEMMAS THEN SOLVE ONE-BY-ONE. IF LACK OF INFRASTRUCTURES, BUILD REQUIRED INFRASTRUCTURES ONE-BY-ONE.

