# OpenReview forum: "AI4SLT: Empirical Processes in Lean 4 for Formal Statistical Learning Theory"
_ICML.cc/2026/Conference — ICML 2026 regular_

### Official Review · Reviewer_ZHqT · 2026-03-11

**Soundness:** 3
**Presentation:** 2
**Significance:** 3
**Originality:** 2
**Overall Recommendation:** 5
**Confidence:** 3

**Summary:**

This paper presents a large-scale Lean 4 formalization (~30,000 lines) of statistical learning theory (SLT) built on empirical process theory. The authors develop formal infrastructure that was previously missing from the Lean ecosystem, including: (1) a complete chain of Gaussian functional inequalities culminating in Gaussian Lipschitz concentration (Efron-Stein $\rightarrow$ Gaussian Poincaré $\rightarrow$ Gaussian LSI $\rightarrow$ concentration), (2) the first formalization of Dudley's entropy integral bound for sub-Gaussian processes via the chaining technique, and (3) an application to least-squares regression (both ordinary and ℓ₁-constrained) achieving sharp minimax rates. The work was carried out through a human-AI collaborative workflow using Claude Code, with humans designing proof strategies and the AI executing tactical proof construction. The authors argue that the formalization process itself exposes implicit assumptions routinely suppressed in textbook proofs.

**Compliance With Llm Reviewing Policy:**

Affirmed.

**Final Justification:**

This paper presents a large-scale Lean 4 formalization of statistical learning theory, covering Gaussian functional inequalities, Dudley's entropy integral via chaining, and least-squares regression applications. Strengths include rigorous end-to-end coverage beyond prior Mathlib scope, a new formalization-oriented proof variant for Dudley (shift-and-Fatou), and a modular design suitable for community reuse.

Original concerns on mathematical novelty, human-AI workflow quantification, compilation metrics, and Mathlib integration were all addressed in the rebuttal. The four-level quantitative breakdown (~15% autonomous, ~30% guided, ~40% specified, ~15% human-critical) and the non-delegable verification observation (three misformalization cases surviving AI self-judging, caught only by human counterexample checking) are contributions to the human-AI collaboration paradigm. Compilation metrics and confirmed Mathlib upstream discussions resolve the presentation and integration concerns. The gap to Bartlett et al. 2020 is characterized. Score raised to accept.

**Key Questions For Authors:**

Q1. Were any textbook proofs found to contain actual errors (not just missing hypotheses) during formalization? If so, this would significantly strengthen the contribution and should be highlighted. If not, please clarify the distinction between "missing assumptions" and "errors."

Q2. Can you provide quantitative data on the human-AI collaboration? Specifically: what fraction of lemmas could the AI agent prove with minimal human guidance vs. requiring significant human proof strategy? What was the typical iteration cycle?

Q3. What is the plan for Mathlib integration? Several components (covering numbers, sub-Gaussian processes, Efron-Stein) seem broadly useful. Has there been any discussion with Mathlib maintainers?

Q4. The paper claims the framework "opens the door" for modern ML theory formalization (double descent, benign overfitting, etc.). Can you concretely assess the gap between your current infrastructure and what would be needed for, say, formalizing Bartlett et al. (2020) on benign overfitting in linear regression?

Q5. The formalization assumes sub-Gaussian processes with continuous sample paths (Theorem 3.8). How restrictive is this in practice for downstream applications, and could the framework accommodate processes with weaker regularity (e.g., separable but not continuous)?

**Limitations:**

The authors acknowledge the scope is limited to the localized empirical process framework and linear/sparse regression applications. However, they do not discuss computational limitations (compilation time, memory usage for large proof terms) or the limits of the human-AI workflow. The societal impact statement is minimal but this seems appropriate for a formalization paper.

**Strengths And Weaknesses:**

**Strengths**
**S1** The scale of 30,000 lines of compiled Lean 4 code with no sorry or unverified axioms is great. The formalization covers an end-to-end pipeline from foundational concentration inequalities through capacity control to concrete regression rates, which goes beyond prior work (Sonoda et al., 2025) that only handled the simpler Rademacher complexity setting. This fills a gap: SLT sits at the intersection of measure theory, functional analysis, and probability, making it harder to formalize than more self-contained mathematical areas.
**S2** The complete formal chain from Efron-Stein through Gaussian Poincaré, the density arguments, Gaussian LSI (via the Rademacher-CLT route and tensorization), to Gaussian Lipschitz concentration is a nice contribution. The density arguments (Theorem 3.3, Lemma 3.4) in particular are routinely hand-waved in textbooks, and making them fully rigorous is valuable.
**S3** The paper makes a compelling case that formalization forces one to confront assumptions that informal proofs suppress. The discussion in Appendix C shows this: the integrability hypotheses and nonemptiness condition in the master error bound are genuine mathematical requirements.
**S4** The three-stage proof (finite nets $\rightarrow$ countable dense sequences via Fatou $\rightarrow$ uncountable sets via continuity) is carefully laid out. The choice of recursive projection over direct projection, yielding tighter constants, is interesting.
**S5** The modular design (covering numbers, chaining, sub-Gaussian processes as standalone components) suggests downstream utility for future formalization efforts in ML theory.

**Weaknesses**

**W1** Limited novelty in the mathematical content. All theorems formalized are well-known classical results (from Boucheron et al., Wainwright, Vershynin). The contribution is purely in the formalization itself. While I believe formalization papers have value, the paper could do more to articulate what new mathematical insights emerged beyond missing hypotheses.

**W2** Human-AI workflow description is superficial. Contribution 4 claims a novel "collaborative formalization paradigm," but Section 1 provides only a high-level sketch (humans design strategies, AI executes). There is no systematic analysis of: which proof steps the AI could handle autonomously vs. where human intervention was critical, failure modes or iteration counts, what fraction of the 500 hours was human vs. machine time, or comparison with fully manual formalization effort.

**W3** No experimental or computational evaluation. The paper provides no data on compilation times, proof term sizes, or comparison of proof lengths against Mathlib conventions. How does the code integrate with or extend Mathlib? Are there plans for upstreaming? The relationship to existing Mathlib infrastructure is discussed only informally.

**W4** Application scope is narrow. The least-squares applications (linear regression, ℓ₁-constrained regression) are relatively elementary applications of the localized empirical process framework. The paper would be strengthened by formalizing at least one more challenging application (e.g., nonparametric regression with Hölder classes, or kernel methods) to stress-test the framework's generality.

**W6** The paper tries to cover a very large amount of material and consequently feels rushed in places. The Lean code snippets, while essential, could be better integrated into the narrative flow. Some design choices (e.g., why use recursive projection for chaining, the two-level entropy integral design with ENNReal/Real wrappers) deserve more discussion in the main text rather than being in appendices. The dependency graph (Figure 2) is helpful but the line counts shown there should be accompanied by a brief discussion of where the most formalization difficulty concentrated.

**W7** The paper does not discuss formalization efforts in adjacent areas of probability theory in other provers (e.g., Isabelle/HOL has substantial measure theory libraries). A comparison of what Lean 4 offers over these alternatives for SLT-specific formalization would strengthen the positioning.

---

> ### Author Rebuttal · Authors · 2026-03-30
>
> We deeply appreciate the reviewer’s efforts and the positive support.
>
> > **W1** *mathematical novelty*
>
> We contribute a new formalization-oriented proof variant for Dudley(Stage 2 in Appendix B): a shift-and-Fatou approach that replaces the standard local-oscillation route. We construct this method because the latter's frequent switching between improper/interval integrals and real/extended-real spaces created persistent refusal preference for AI.
>
> ---
> > **Q1** *missing assumption vs. errors*
>
> We do not find outright logical errors. By "missing assumptions" we mean hypotheses that are mathematically necessary and silently used but never explicitly stated. These are not errors because the theorems remain true under the intended (but unstated) conditions. The distinction is that an error would make a statement false, whereas a missing assumption makes it incomplete.
>
> ---
>
> > **W2 & Q2** *Human-AI Workflow, comparison with manual.*
>
> We invite reviewer to our reply for Reviewer `7kuu` on details of workflow with systematic analysis. Furthermore, our codebase (34,640 lines) is approximately 3.6× larger than the fully manual [lean-rademacher](https://github.com/auto-res/lean-rademacher.git) project (~9,500 lines, 5 contributors), which added ~4,200 lines over 10 months of incremental development.
>
> ---
>
> > **W3 & Q3** *Compilation time, proof complexity against Mathlib, relation to Mathlib, upstream plan*
>
> We provide quantitative data below, which we will incorporate in the revised manuscript.
>
> **Compilation and proof term metrics.** Our SLT library (contains zero sorry & axiom, across 34,640 lines, 55 files) compiles fresh in 743±44s on 1 consumer CPU with 16 GB RAM (peak usage ~3 GB) over 3 runs, producing 43.3 MB of verified proof terms.
>
> **Proof complexity.** Direct proof-length comparison is not meaningful since our contribution is precisely formalizing results ***beyond Mathlib's frontier***. The overlap exists (Chernoff bounds, variance, Gaussian integrals), we import them from Mathlib directly rather than reproving.
>
> **Relation w. Mathlib.** Tactic-level analysis reveals that 71.8% of premise references resolve to Mathlib, invoking 1,668 unique Mathlib lemmas/definitions. The remaining 19.7% are project-internal and 8.5% are Lean core. The project imports 130 distinct Mathlib modules. We will add a metrics in the revision.
>
> **Mathlib integration.** We have discussed with one Mathlib maintainer on the integration plan and we plan to sequentially upstream our library to Mathlib.
>
> ---
>
> > **W4** *application scope*
>
> We agree that a broader application would better showcase the framework’s scope. Our main contribution is the reusable formal program as a workhorse, while linear and $\ell_1$-constrained regression serve as clean end-to-end validations. We leave the mordern applications as future work.
>
> ---
>
> > **Q4** *benign overfitting*
>
> While our infrastructure covers concentration inequalities, metric entropy, and covering number calculus, formalizing Bartlett et al. (2020) would additionally require a random matrix theory stack and spectral perturbation theory. None of these exist in Mathlib, representing a significant and currently unbridged formal gap.
>
> ---
>
> > **Q5** *path-continuity*
>
> In the downstream applications, after passing to the localized empirical image, the process is linear in index and hence continuous so this can cover richer classes such as nonparametric classes, RKHS/kernel classes, or neural networks. A separable-but-not-continuous version should also be possible in principle by replacing path continuity with a suitable measurable separability/modification assumption (overkill for SLT cases).
>
> ---
>
> > **W6** *paper presentation*
>
> Thank you for the helpful suggestion. In the revision, we will tighten the main-text presentation by integrating the Lean snippets more directly into the surrounding exposition, moving the motivation for key design choices from the appendix into the main text.
>
> Furthermore, we decompose the library using LeanDojo-v2, extracting 865 traced theorems comprising 18,669 total tactic steps (avg 21.6, 90th percentile 51). The most tactic-intensive files — `Dudley.lean` (1,661 steps, avg 237/theorem), `EfronStein.lean` (1,204 steps, avg 67/theorem), `GaussianLipConcen.lean` (993 steps, avg 43/theorem) — all correspond to the paper's core technical contributions and where difficulty concentrates. We will add this discussion after Fig. 2 as reviewer suggested.
>
> ---
>
> > **W7** *comparison w. other proof assistants*
>
> While Isabelle/HOL has formalized basic concentration inequalities (Markov, Chebyshev, McDiarmid, Efron–Stein) and Rocq has covered Hoeffding/Bernstein, none of these efforts include the advanced Gaussian analysis pipeline or Dudley's entropy integral that our work provides. Moreover, Lean 4's Mathlib offers a rapidly growing measure-theoretic and analysis stack with strong community support. We will add discussion & comparison in the revised manuscript.

---

> > ### Author Rebuttal · Reviewer_ZHqT · 2026-04-04
> >
> > Given the rebuttal, including the response to Reviewer 7kuu on the human-AI collaborative workflow, my concerns are fully resolved. The quantitative four-level breakdown (~15% autonomous, ~30% guided, ~40% specified, ~15% human-critical) answers W2/Q2. The non-delegable verification observation, that three mathematical misformalization cases survived AI self-judging and were only caught by human counterexample checking, is a finding that strengthens Contribution 4. The compilation metrics (743$\pm$44s, 43.3 MB proof terms, 71.8% Mathlib premise resolution) from W3/Q3 resolve my presentation concern. The gap analysis to Bartlett et al. 2020 in Q4 is characterized. The authors have also committed to moving the quantitative breakdown table to the main body, addressing Reviewer 7F7R's presentation concern. I am raising my score accordingly.

---

> > > ### Author Response · Authors · 2026-04-08
> > >
> > > Thanks for your effort and time.

---

### Official Review · Reviewer_7kuu · 2026-03-12

**Soundness:** 3
**Presentation:** 4
**Significance:** 4
**Originality:** 4
**Overall Recommendation:** 5
**Confidence:** 3

**Summary:**

This paper presents a large-scale Lean 4 formalization of statistical learning theory grounded in empirical process theory. The contributions include formalized proofs of Gaussian Lipschitz concentration , Dudley's entropy integral bound for sub-Gaussian processes, and an application to least-squares regression achieving sharp minimax rates. The project comprises approximately 30,000 lines of Lean 4 code developed through human-AI collaboration (~500 hours).

**Compliance With Llm Reviewing Policy:**

Affirmed.

**Key Questions For Authors:**

Could you provide more quantitative details on the human-AI collaboration — e.g., what percentage of lemmas were completed by the AI on first attempt vs. requiring multiple iterations?

**Limitations:**

See above weaknesses.

**Strengths And Weaknesses:**

Strengths:

The scope and ambition of this work are impressive. Formalizing the full pipeline from concentration inequalities through chaining arguments to concrete regression rates is a genuinely substantial undertaking.

The human-AI collaborative workflow is a practical contribution in its own right. Demonstrating that ~500 hours of supervised development can produce 30,000 lines of verified Lean code for a technically demanding area suggests a viable model for future formalization projects.

Beyond verification, this work could also open the door for automated theorem proving (ATP) approaches in machine learning theory. By providing a structured, machine-readable library of SLT results in Lean 4, the formalization creates a foundation that ATP systems could build upon — for example, to automatically discover or verify new generalization bounds.

The presentation is also great.

Weaknesses:
The paper would benefit from a more honest discussion of the limitations of the current formalization scope. In particular, how far is this framework from being applicable to a real use case such as analyzing neural networks? The application section covers linear regression and ℓ₁-constrained regression, which are relatively classical settings. A concrete assessment of the gap between the current infrastructure and what would be needed for modern architectures would strengthen the paper.

The discussion of the human-AI workflow is somewhat superficial. The paper states that humans design proof strategies and AI executes tactical proof construction, but provides little detail on failure modes, iteration cycles, or what fraction of AI-generated proofs required significant human correction. A sample prompt-to-proof workflow showing how a specific lemma was decomposed, delegated to the AI, and refined would be interesting and instructive for the community.

---

> ### Author Rebuttal · Authors · 2026-03-30
>
> We deeply apprecaite reviewer's effort and constructive comments on this work.
>
> > **Can use for neural networks?**
>
> Our framework is not yet directly applicable to modern architectures such as neural networks; however, the underlying proof pipeline is largely aligned and already provides the core tools and infrastructure that would be required in such settings. For example, the pipeline `RegressionModel` and lemmas `local_gaussian_complexity_bound` and  `master_error_bound` can be used for over-parameterized neural networks [1] with covering lemma for convex hulls.
>
> > **Human-AI Collaborative Workflow**
>
> We present three transferable methodological contributions and one critical observation of human-AI formalization, supported by ~60 preserved specifications. We also classify the declarations in our library into four levels (human intervention vs. AI automation) and provide a quantitative breakdown.
>
> **Contribution 1: Structured Specification Protocol.** Unstructured prompts are a major bottleneck in human–AI formalization, failing roughly **70% of the time** in our setting. We address this by introducing a structured `TASK.md` format consisting of four parts: the exact target statement, infrastructure pointers (e.g., lemma names and file paths), a formalization-oriented step-by-step proof plan, and hard constraints such as forbidding `sorry` and new axioms. **This reduces first-attempt failure to roughly 15%**. In addition, every component completed within two iterations relied on a complete `TASK.md`. As one example, the Gaussian Poincaré proof was split into four such tasks, each of which compiled on the first or second attempt.
>
> **Contribution 2: Specification Evolution.** Retrying with the same spec never works, each failure demands a more precise spec resolving a specific ambiguity. We address this by introducing an iterative-evolved revision framework by AI's feedback. For example, the GLSI density extension required four specs:
> - the first hand-waves entropy lower-semicontinuity as "Fatou-type arguments" — the AI failed because Lean's Fatou demands nonneg integrands while $t \log t$ goes negative.
> - The second adds infrastructure pointers but left negativity unresolved.
> - The third decomposes the problem, factoring out the gradient convergence piece.
> - The fourth closes the argument with three precise additions: the shift trick for nonnegativity, the computation domain ("work in `ENNReal`"), and the exact Mathlib lemma name.
>
> Each iteration resolved a distinct ambiguity class: vague strategy → dependencies → decomposition → trick, domain, and API. The AI's failures are informative — they reveal what the human left implicit, making collaboration **convergent**.
>
> **Observation: Non-Delegable Verification.** We identify the most dangerous failure: statements that type-check but are mathematically false. E.g., we identify three misformalization survived from AI's self-judge: a false gap bound requiring control of `N(ε)/N(2ε)` not implied by `TotallyBounded`, an impossible disjunction false for specific parameters, and a Riemann rectangle misused as an upper bound. The AI reported `sorry` and kept trying — only the human, via counterexamples, detected the errors. We recommend: human review of every (at least major/target) statement before proving, counterexample checking for inequalities, and treating persistent `sorry` as a signal to verify the statement, not just the proof.
>
> **Quantitative Breakdown.** By tracing formalized lemmas/theorems by human-authored specifications, we compute the fractions of workloads in four levels.
>
> | Level | Description | Fraction |
> |-------|-------------|----------|
> | **Autonomous** | AI proves from statement alone | ~15% |
> | **Guided** | Human provides proof sketch; AI fills details | ~30% |
> | **Specified** | Human writes complete NL proof; AI translates to Lean 4 | ~40% |
> | **Human-critical** | Human must intervene after AI failure (false statement, wrong approach) | ~15% |
>
> > a sample prompt-to-proof of Human-AI workflow
>
> We thank the reviewer for this suggestion and provide the original evolved specifications for the GLSI example in **Contribution 2** in [anonymized repository](https://anonymous.4open.science/r/lean-stat-learning-theory-prompt-evolve-E4FD/), which will be incorporated into the revised manuscript and released with the full codebase.
>
> ---
>
> **Reference**
>
> [1] Learning with Norm Constrained, Over-parameterized, Two-layer Neural Networks. JMLR 2024.

---

> > ### Author Rebuttal · Reviewer_7kuu · 2026-04-03
> >
> > Thank you for your response.

---

> > > ### Author Response · Authors · 2026-04-08
> > >
> > > Much appreciated for your effort and time.

---

### Official Review · Reviewer_2a5M · 2026-03-19

**Soundness:** 3
**Presentation:** 2
**Significance:** 3
**Originality:** 3
**Overall Recommendation:** 4
**Confidence:** 3

**Summary:**

This paper presents a large-scale formalization of foundational results in statistical learning theory using the Lean 4 theorem prover.

**Compliance With Llm Reviewing Policy:**

Affirmed.

**Final Justification:**

The rebuttal has addressed my concerns. So I raise my score to 4 after rebuttal.

**Key Questions For Authors:**

Could you give a concrete example how to use the Lean-code to prove?

**Limitations:**

yes

**Strengths And Weaknesses:**

Strengths:   It is significant to Lean 4 forformalization of statistical learning theory (SLT) grounded in empirical process theory.

Weakness: (1) The paper is quite hard to understand because it tries to cover a broad topic without dealing with the crucial details.  The goal from Statistical Learning theory to Lean 4 looks very promising but the formalization in Section 3 looks quite scattered and is not well-organized.

(2) The paper provides no evidence that the Lean4 code actually compiles or that the theorems are correctly proved.

（3） The paper does not provide examples how to use the Lean 4-code to show some other statements in SLT.  So we don’t see any empirical evidence of the significance of this paper.

 (4) It seems that the paper concludes in a rush, which leaves an impression that it is not ready for publication.

---

> ### Author Rebuttal · Authors · 2026-03-30
>
> We thank the reviewer for the thoughtful comments. We address your concerns as below.
>
> > **W1** *presentation/organization*
>
> Section 3 is organized by the dependency graph in Fig. 2: **The Gaussian-analysis line** proceeds from Efron–Stein to Gaussian Poincaré to density arguments to Gaussian LSI to Gaussian Lipschitz concentration, while **the empirical-process line** proceeds from covering numbers to chaining to metric entropy to Dudley.
> Section 4 then combines these components in the least-squares framework.
>
> Each natural-language theorem is followed by human-verified Lean statement precisely to make the formal interfaces and hidden assumptions explicit. We agree that this structure can be signposted better, and in the revision we will add a short roadmap paragraph at the start of Section 3 and move part of the proof-engineering details to the appendix.
>
> >**W2** *correctness/compilation*
>
> In Lean, the proof certificate is kernel checking. A theorem is not merely “claimed”, instead it is accepted only if the corresponding proof term compiles in the trusted kernel. Our library compiles without `sorry` or additional `axiom`. Appendix A lists the major formalized results with exact textbook references. We agree that this point should have been made more explicit for non-Lean readers, and we will add a brief artifact paragraph describing the build/verification procedure.
>
> >**W3** *how to use code to show statements in SLT*
>
> Sections 4.1–4.2 have already served exactly this purpose: starting from the core infrastructure (concentration, Dudley, covering numbers), we formalize sharp rates for both linear regression and $\ell_{1}$-constrained regression by importing and composing these tools, demonstrating that the library is directly reusable for deriving new SLT results with substantially less effort than building from scratch.
>
> >**W4** *readiness/conclusion*
>
> We acknowledge that the transition from the application section to the conclusion feels abrupt due to the page limit. We will detail Section 4 with more discussion of the formalization strategy and challenges in the least-squares framework, providing a smoother narrative arc before the conclusion.
>
> ---
>
> We hope our feedback address your concerns and appreciate your time and efforts.

---

> > ### Author Rebuttal · Reviewer_2a5M · 2026-04-08
> >
> > I am satisfied with the rebuttal.

---

> > > ### Author Response · Authors · 2026-04-08
> > >
> > > Thanks for your effort and time.

---

### Official Review · Reviewer_7F7R · 2026-03-22

**Soundness:** 4
**Presentation:** 2
**Significance:** 3
**Originality:** 2
**Overall Recommendation:** 4
**Confidence:** 4

**Summary:**

This paper provides the first Lean 4 formalization of a few foundational results in statistical learning theory, including some high-dimensional probability toolbox and Dudley's entropy integral. They apply this to formalize some results in nonparametric regression. They achieved this through an extensive human-AI collaboration.

**Compliance With Llm Reviewing Policy:**

Affirmed.

**Final Justification:**

Changes in the scores:
- Presentation: 3->2
- Significance: 2->3
- Overall: 2->4

The current version of the paper is primarily focused on explaining the structure of their formalization. As such, I initially tried to evaluate the significance of their formalization library itself, which didn't really appeal to me. However, in the authors' rebuttal, it was noted that the ML theory formalization project they are working towards is meant to be the library used by the future AI to automatically verify the sophisticated proofs and calculations often present in theory papers. This does seem to be an important and exciting goal, which would potentially benefit the theory community. The authors also resolved my concern about the difficulty of building such a formalization library with the assistance of AI, by showing in the rebuttal that the AI alone couldn't really do much of the work and human contributions were essential.

That said, the presentation of the current manuscript makes it difficult to evaluate certain points made in the rebuttal. As noted in my acknowledgement, researchers trying to build on their work might find many details missing when trying to adapt their human-AI collaboration protocol. The authors' rebuttal to the reviewer 7kuu was very informative, but I really wish they were in the main body with further workflow description attached in the appendix or the supplement. They also emphasized the novelty of their new proof of Dudley's theorem, but I don't find it mathematically interesting and it's hard to assess the usefulness of their new approach in proof formalization in general (though I acknowledge some merits).

In conclusion, I am now leaning towards acceptance, given that the paper undergoes a careful revision. It wouldn't be a surprise if it gets rejected though.

**Key Questions For Authors:**

None

**Limitations:**

Yes

**Strengths And Weaknesses:**

- Soundness: As a paper dealing with the formalization of known results, there is no issue in soundness.

- Presentation: The authors presented their results in a clear way, along with nice figures that summarize their formalization steps. The theorems and lemmata leading to the results targeted for formalization were easy to follow. They also provided some Lean 4 codes in their main body and the appendices, but the codes were not very readable and wouldn't be very helpful for general audience. It would be better to reformat the codes, add comments, or simply remove them (since they are given in supplements).

- Significance: Theoretical results the authors formalized are indeed vital in machine learning theory and are fundamental in rigorous justifications and guarantees of machine learning and statistical methods. However, at the moment, it is generally unlikely that the researchers would put significant effort on writing Lean codes to formalize their argument and the reviewers really care about the machine-verified correctness, unless the results are truly surprising and/or counterintuitive. As human-AI collaboration grows in the future, it might become easier to produce AI-generated formalization and so people care more about them, but that again questions the significance of the present work if the AI can do them so easy. Consequently, their work might appeal to the proof formalization/Lean community, but I am skeptical of their significance from a viewpoint of artificial intelligence/machine learning researchers. On the other hand, it would have been more interesting if an efficient human-AI collaboration paradigm for proof formalization were described in this paper in greater detail.

- Originality: This paper presents some original contributions in that they provided the first proof formalization in the area of statistical learning theory, but not in a mathematical or theoretical sense. It would be nice if a clean formalization requires a new proof or new perspectives on the existing results, but the authors did not highlight any and it doesn't seem so. The authors claim that their work is not merely an implementation and demands comprehensive understanding of the theory, which I agree, but that alone does not add to the theoretical originality of their work.

---

> ### Author Rebuttal · Authors · 2026-03-30
>
> We apprecaite reviewer's effort and constructive comments. In the revision, we will improve the readability of Lean snippets with better formatting also move more of them to the appendix. In the next, we will clarify the significance and novelty for better understanding.
>
> > *Contribution on Human-AI workflow*
>
> We introduce a structured specification protocol that reduces AI first-attempt failure **from ~70% to ~15%**, and provide a quantitative breakdown showing only ~15% of lemmas were provable autonomously. Details on this workflow are in our reply to Reviewer `7kuu`.
>
> > *Significance for ML theory formalization*
>
> We respectfully argue that formalizing such results in Lean is not merely a one-time verification effort, but rather the construction of reusable infrastructure that can be incrementally extended by the community. Such **infrastructure** has its own value in the era of "auto-research".
>
> ML theorists do not need to write Lean. Instead, in the future, ML theory papers can be automatically attached with Lean proofs (by AI systems) for correctness validation, thereby providing guarantees that go beyond traditional peer review.
>
> > *future AI does not diminish the contribution*
>
> The reviewer suggests AI "in the future" will make formalization "so easy" that our work becomes moot. We respectfully disagree that this confuses *using* a library with *building* one. AI-assisted formalization improves as verified libraries grow, because these libraries serve as both training data and callable **infrastructure**.
>
> Our open-sourced library is exactly such a resource. Without it, future AI has neither the data to learn from nor the lemmas to invoke. The current state confirms this: Opus-4.5 could prove only ~15% of lemmas autonomously, with the deep dependency chains remaining far beyond its capability.
>
> > *Theoretical/ Mathematical novelty in ML theory*
>
> We understand the reviewer's concern on mathematical countribution in the perspective of ML theory itself. Though not our main target, we still have the mathematical contribution to ML theory: we provide a **new proof of Dudley’s entropy integral theorem**, which is **pedagogically clearer** and **more convienient for formulation** when compared to textbook proof, see Appendix B for details.
>
> To be specific, the chaining argument from textbook is classical, but the Stage-2 passage from finite good dyadic nets to a countable dense index set is handled by a shift-and-Fatou argument, which is different to standard textbook routes. Our proof is constructed using feedbacks from agent's formalization process. We start with the common-textbook local oscillation strategy, but observe that the agent has confusion with choosing:
>    - improper integral vs. interval integral, and
>    - real space vs. extended real space.
>
> This strategy switches frequently between these types, creating a hard burden for formalization and resulting a high refusal preference for agent, i.e. the agent persistantly writes formal claim using `sorry` placeholder and no progress. Therefore, we construct the shift-and-Fatou argument to clearly separate into two classes and only need to convert to improper integral on extended real then back interval integral on real for once.
>
> Beyond formalization, this proof framework is also pedagogically clearer: it isolates the type-conversion bookkeeping into a single explicit step, letting readers follow the core probabilistic intuition (finite nets → countable → uncountable) without the measure-theoretic distractions that obscure the standard textbook route.
>
> This is relevant not only to the Lean community, but also benefit to the broader ML theory community. We hope this clarification would address the reviewer's concerns.
>
> ---
>
> We hope our feedback would clairfy misunderstanding and we are happy to address any potential concerns.

---

> > ### Author Rebuttal · Reviewer_7F7R · 2026-03-31
> >
> > Thank you for your clarification. It seems like your main contribution is towards building an extensive formalization library for AIs to automatically verify theoretical works in AI/ML in the future. This sounds promising and adds to the significance of the your work, but then I believe the human-AI collaboration paradigm in proof formalization should indeed be the main focus of the paper. The current manuscript mostly explains the overall structure and the implementation challenges of the formalization, but people would generally be happy with Figures 1 and 2 and much of those details could be moved to the appendix. Instead, I think the following should be emphasized more in the main body:
> >
> > - the table you presented in the rebuttal below
> > - in what part of the formalization did AI struggle/excel?
> > - what part needed heavy human intervention?
> > - did AI manage to formalize some results with your library in hand that otherwise would not have succeeded?
> > - proof strategy that deviates from textbook approaches (additional question: how novel is the new proof? Is the proof idea completely new, or is it a simple tweak for the ease of formalization?)
> > - how to design an efficient human-AI collaboration strategy? (e.g. can your strategy be somehow universally adapted to formalizing other results?)
> > - are there any reusable (sub-)components of your formalization useful for future work?
> >
> > If I were someone who seeks to attach automated AI-assisted verification to my theory or to build on your work and expand the AI/ML theory formalization infrastructure, these details would be the most helpful, which seem to be missing in the current revision. The paper would be of more publishable quality if the above concerns are resolved.

---

> > > ### Author Response · Authors · 2026-04-01
> > >
> > > Thank you for the constructive follow-up and for acknowledging our contributions to both the formulation and the human–AI system. We would like to clarify that, before emphasizing the human–AI system, we need to introduce and detail the formulation/library first. For example, Figures 1 and 2 provide helpful overviews of the dependency structure, but are not sufficient. Future users will need the reusable interfaces and the precise side conditions in order to build on this work effectively.
> > >
> > > This organization will give readers a clear understanding of what we have done, and then how we can do in the future via such a system. Following the reviewer’s suggestion, we will provide more details about the system in both the main text and the appendix with the following:
> > >
> > > 1. **The quantitative breakdown table.** The table in reply to Reviewer `7kuu` will be added to the main body.
> > >
> > > 2. **Where AI excelled vs. struggled & Heavy human intervention.** We will use examples to illustrate:
> > >    - AI excelled at routine measure-theoretic bookkeeping (e.g., integrability, measurability) and at filling in tactic-level given a complete NL proof.
> > >    - AI struggled with deep dependency chains grounded from Mathlib, novel proof constructions, and mismatched formal statements which are type-checked. Those require heavy human intervention.
> > >
> > > 3. **The library changes what AI could formalize.** This is precisely why we view the library as the first-order contribution. As Figures 1 & 2 show, downstream results depend on a deep chain of parent infrastructure. Before that infrastructure existed, the agent had no way to build downstream results — the needed interfaces were missing. Once in place, the least-squares results (Section 4) were completed by direct invocation.
> > >
> > > 4. **Novelty of the new proof.** The novelty is the application of shift-and-Fatou argument to Dudley's theorem, which overcomes the frequent switching between improper/interval integrals and ℝ/ℝ≥0∞ in the standard textbook route. No textbook clearly states this before.
> > >
> > > 5. **Transferability of the strategy.** Our protocol and the specification evolution framework are domain-agnostic, which can be adapted to any formalization project where a human expert decomposes problems for an AI agent.
> > >
> > > 6. **Reusable components.** The library is designed for modularity. The covering number infrastructure, sub-Gaussian process, the Gaussian analysis toolbox (Efron–Stein, Poincaré, LSI, density arguments), and the empirical process framework can be directly imported and used by both human researcher and AI.
> > >
> > > ---
> > >
> > > We hope our clarification fully address your concerns and appreciate if you could reconsider your evaluation.

---

### Decision · Program_Chairs · 2026-04-30

**Decision:**

Accept (regular)

**Comment:**

This paper formalizes some classical results related to statistical learning theory in Lean, a
popular formal proof assistant. While these results are not mathematically novel, the
formalization of them without the use of AI would have been a substantially difficult task.
In this paper, the authors discuss how they were able to make progress using human-AI collaboration.
After back-and-forth with the reviewers, the authors have
proposed to edit the next version of their paper in a way which should make the key
observations/contributions related to the Human-AI workflow more accessible to future researchers.
Because I think this aspect may be of interest to other researchers in AI for math and Human-AI collaboration,
I recommend acceptance.